# CannyEdit: Selective Canny Control and Dual-Prompt Guidance for Training-free Image Editing

## Abstract

Recent advances in text-to-image (T2I) models have enabled training-free regional image editing by leveraging the generative priors of foundation models. However, existing methods struggle to balance text adherence in edited regions, context fidelity in unedited areas, and seamless integration of edits. We introduce ***CannyEdit***, a novel training-free framework that addresses this trilemma through two key innovations. First, *Selective Canny Control* applies structural guidance from a Canny ControlNet only to the unedited regions, preserving the original image's details while allowing for precise, text-driven changes in the specified editable area. Second, *Dual-Prompt Guidance* utilizes both a local prompt for the specific edit and a global prompt for overall scene coherence. Through this synergistic approach, these components enable controllable local editing for object addition, replacement, and removal, achieving a superior trade-off among text adherence, context fidelity, and editing seamlessness compared to current region-based methods. Beyond this, CannyEdit offers exceptional flexibility: *it operates effectively with rough masks or even single-point hints in addition tasks*. Furthermore, the framework can seamlessly integrate with vision-language models *in a training-free manner* for complex instruction-based editing that requires planning and reasoning. Our extensive evaluations demonstrate CannyEdit's strong performance against leading instruction-based editors in complex object addition scenarios.

**(a) Addition in a complex scene:** Add ***another woman*** running on the treadmill.

**(b) Dual additions requiring planning:** Add a dog lying on the ground and add a woman reading ***on the sofa***.

**(c) Reasoning-based edit:** Make the food be healthy (VLM's reasoning: replace the fried food with healthy food and add a salad bowl).

**(d) Edit with user-provided mask:** Add a ***small single-person*** sofa in the middle of the home.

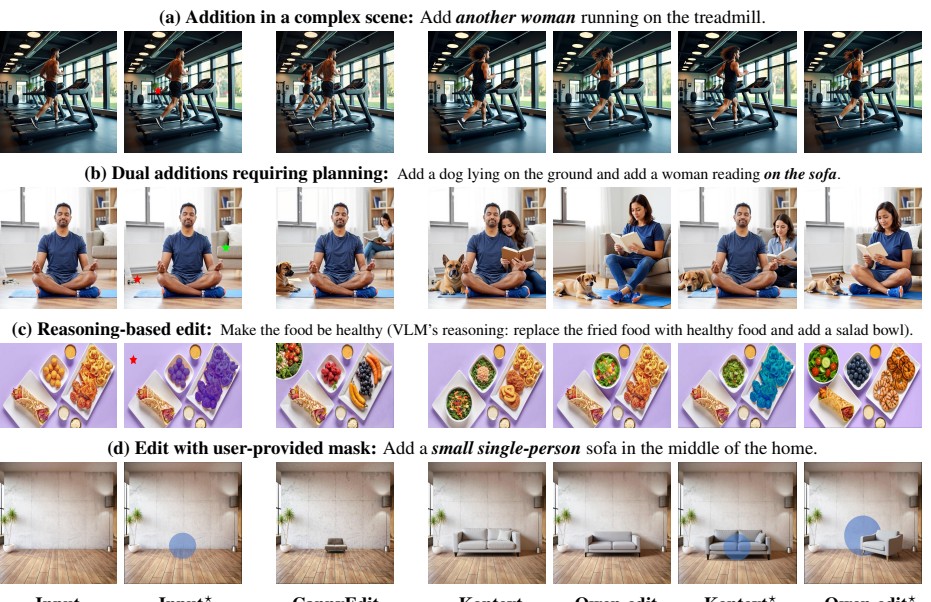

| **Input** | **Input**$^\star$ | **CannyEdit** | **Kontext** | **Qwen-edit** | **Kontext**$^\star$ | **Qwen-edit**$^\star$ |

Figure 1: CannyEdit, a training-free method, demonstrates versatility by functioning as both a precise mask-based editor (d) and a flexible instruction-based tool (a, b, c) with point hints for edits provided by vision-language models (VLMs). Its training-free framework enables seamless integration with leading VLMs to handle complex edits requiring planning or abstract reasoning. In contrast, competitors like FLUX.1 Kontext [dev] (Kontext) (Labs et al., 2025) and Qwen-Image-Edit (Qwen-edit) (Team, 2025b) struggle with these tasks, even when provided with identical visual hints. Inputs and competitors' results using these hints are marked with $^\star$.

# 1 INTRODUCTION

Recent advances in text-to-image (T2I) models have achieved substantial progress in quality and controllability (Rombach et al., 2022; Betker et al., 2023; Chen et al., 2023; Esser et al., 2024; Black-Forest Labs, 2024). These advancements have enabled diverse downstream applications, utilizing their enhanced quality, efficiency, and versatility. In this work, we address one of the most challenging applications: *region-based image editing*, which entails modifying user-specified image areas (e.g., adding, replacing, or removing objects) while maintaining overall image consistency. Region-based image editing extends standard T2I generation by introducing a critical constraint: the generated content must align not only with the text prompt (***Text Adherence***) but also with the existing visual context of the image (***Context Fidelity***). This problem is commonly known as the *editability-fidelity trade-off*.

Initial efforts of region-based editing centered on training-based inpainting methods (Rombach et al., 2022; Zhang et al., 2023a; Zhuang et al., 2024; Black-Forest Labs, 2024), which train a model to fill masked regions from a text prompt. While being good at maintaining the context fidelity, these approaches are often sensitive to mask shape and struggle to generalize to realistic interactions, affecting the adherence to instructions to generate high-quality desired edits. More recently, the research of training-based methods has shifted to instruction-based models (Brooks et al., 2023; Li et al., 2024d; Hui et al., 2024; Wei et al., 2024; Liu et al., 2025; Labs et al., 2025; Wu et al., 2025; Deng et al., 2025; Team, 2025b), which excel at various free-form edits. Their key limitation, however, is a lack of precise spatial control; as shown in Figures 1 (d), and 13, even leading models can fail to reliably target user-specified areas to edit despite clear guidance.

In this paper, we explore an alternative: training-free methods that leverage the generative priors of foundation T2I models. While initially developed for UNet-based diffusion models (Hertz et al., 2022; Cao et al., 2023; Tumanyan et al., 2023), recent work has shifted to more advanced rectified-flow-based Multi-Modal Diffusion Transformers (MM-DiTs) (Rout et al., 2024; Wang et al., 2024; Deng et al., 2024; Tewel et al., 2025; Zhu et al., 2025). A key advantage of MM-DiTs is their flexibility to control the generation process, e.g., one can inject the query/key/value of source-image tokens (obtained via inversion (Deng et al., 2024; Rout et al., 2024; Wang et al., 2024)) into that of the generated tokens at each denoising timestep, improving context fidelity. However, the improved context fidelity often comes at a cost of text adherence, as exemplified by results of RFSolver-Edit (Wang et al., 2024) under different injection steps in Figure 3 (b.1-b.6), where it is unable to strike a good balance between the two criteria. A quantitative study on this is shown in Figure 2.

To achieve a better trade-off, KV-Edit (Zhu et al., 2025) introduces user-provided masks to separate regions to be edited from those to be preserved. During generation, KV-Edit only updates the image tokens in the unmasked regions while keeping the masked regions intact. Although this greatly improves the trade-off (as shown by the orange points in Figure 2), KV-Edit sometimes produces conspicuous artifacts and inconsistencies at mask boundaries, especially when the mask is not precise. Typical examples are shown in Figure 3 (c.1, d.1). This imperfection highlights another key aspect of region-based image editing: ***Editing Seamlessness***, which is essential to good user experience. We hypothesize that the boundary artifacts of KV-Edit are sourced from its hard context replacement strategy, which ensures context fidelity but breaks the interdependency necessary for smooth boundary transitions.

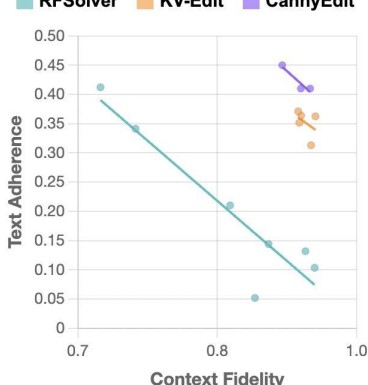

Figure 2: Quantitative study shows that CannyEdit achieves the best editability-fidelity trade-off under varying hyperparameter settings.

Building upon these insights, we introduce *CannyEdit*, a novel training-free image editing method designed to resolve the core trilemma of editability, fidelity, and seamlessness. CannyEdit is built on two synergistic innovations that directly address the shortcomings of prior work.

To overcome the harsh artifacts of rigid masking, we first propose ***Selective Canny Control***. This "soft" control strategy uses a Canny ControlNet (Zhang et al., 2023b) to enforce structural guidance from the source image only on unedited regions. This preserves the original layout at the unedited

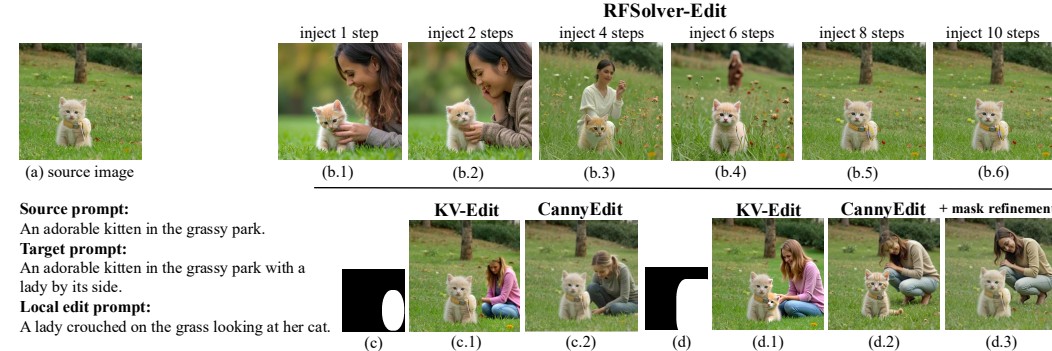

Figure 3: **Editing results of different methods.** RFSolver-Edit's outputs fail to balance context fidelity and successful addition simultaneously. While KV-Edit and CannyEdit deliver a better trade-off, CannyEdit results in more natural and seamless edits whereas KV-Edit's results introduce artifacts such as a partially cropped subject in (c.1), and a partially generated extra cat in (d.1).

region while creating a structurally flexible canvas for the edit. It avoids the partition of edited and unedited regions on the latent noise directly. To ensure the generated content is both accurate and contextually coherent, we then introduce ***Dual-Prompt Guidance***. This technique uses a specific *local prompt* for high-precision text adherence to editing instructions, while a global *target prompt* maintains overall scene consistency and facilitates realistic interactions. This dual-prompt control is achieved by modifying attention computations within the MM-DiTs. Together, these two techniques ensure a smooth and natural integration of edits, as examples shown in Figure 3 (c.2, d.2).

This powerful combination allows CannyEdit to achieve **a superior trade-off among text adherence, context fidelity, and editing seamlessness** compared to region-based baselines like KV-Edit (Zhu et al., 2025) and popular inpainting methods such as BrushEdit (Li et al., 2024d), PowerPaint (Zhuang et al., 2024), and FLUX Fill (Black-Forest Labs, 2024). Moreover, it achieves a significant qualitative leap in editing realism, as validated by a user study and VLM-based evaluation.

Crucially, CannyEdit's training-free and flexible framework unlocks capabilities beyond traditional region-based editing:

- *Multi-Region Editing:* It can perform multiple distinct edits in a single generation pass.
- *Flexible Guidance:* It operates effectively with imprecise spatial cues like rough masks or single-point hints, preserving context fidelity (Figure 3, (d.2, d.3)).
- *Zero-Shot VLM Integration:* It pairs a VLM for high-level reasoning with CannyEdit for precise execution, enabling complex edits that require planning (Figure 1, (b)(c)).

We demonstrate that in a controlled setting for complex object addition, **CannyEdit quantitatively outperforms leading open-sourced instruction-based editors** like FLUX.1 Kontext [dev] (Labs et al., 2025) and Qwen-image-edit (Team, 2025b) in context fidelity and text adherence when all methods are provided the same VLM-inferred point hints.

## 2 RELATED WORK

**Training-based image editing methods.** Training-based methods can be categorized into three streams. First, *inpainting* trains models to fill masked regions based on text prompts (Rombach et al., 2022; Zhang et al., 2023a; Zhuang et al., 2024; Black-Forest Labs, 2024). However, it is sensitive to mask shapes, and often exhibits weak text adherence along with boundary artifacts. In contrast, CannyEdit's training-free design leverages the generalization of foundational T2I models; its soft structural control enhances seamlessness and, by refining the edit region during generation, reduces the dependence on precise masks. Second, *instruction-based models* are trained on large-scale before-and-after image pairs combined with instruction prompts (Liu et al., 2025; Labs et al., 2025; Wu et al., 2025; Deng et al., 2025; Team, 2025b). These models enable impressive free-form edits but lack robust spatial controllability. CannyEdit addresses this limitation by incorporating explicit, mask-driven localization and scale control. Third, *co-training editors with VLMs* aims to enhance editing reasoning abilities (Fu et al., 2024; Huang et al., 2024; Fang et al., 2025; Zhou et al.,

2025). However, this approach requires extensive training and locks the models to specific VLMs. In contrast, CannyEdit can integrate with VLMs in a training-free manner, allowing for on-the-fly upgrades without the rigidity and high costs associated with co-trained systems.

**Training-free image editing methods.** Training-free editing methods typically use a two-stage attention-based pipeline: (1) Inversion: the source image is inverted into the diffusion model's latent space to extract attention features during each denoising step; (2) Source-attention injection: these features are injected into text-image cross-attention during target image generation. Early works like Prompt-to-Prompt (Hertz et al., 2022) and MasaCtrl (Cao et al., 2023) manipulated cross-attention maps in the UNet model between text tokens and image regions. Recent methods (Rout et al., 2024; Wang et al., 2024; Deng et al., 2024; Tewel et al., 2025; Zhu et al., 2025) extend this to more advanced rectified-flow-based transformer models, such as FLUX. As discussed in Section 1, balancing text adherence in edited regions, preserving unedited context, and achieving seamless integration remain key challenges for existing methods. In contrast to approaches that rely solely on attention injection, our work introduces a novel framework that combines explicit structural control with a modified attention process for dual-prompt guidance.

**Controllability in T2I generation**. Introducing controllability in text-to-image (T2I) generation significantly advances precise image manipulation, unlocking various applications (Tan et al., 2024; Zhang et al., 2023b; Li et al., 2024a;c; Zhao et al., 2023). ControlNet is a pivotal framework integrating multiple conditional inputs—such as Canny edge maps for layout and depth maps for spatial arrangement. Specifically, we identify Canny ControlNet as effective for image editing. As an optional plug-and-play module for foundational T2I models, it ensures generated images adhere to both textual prompts and structural details from Canny edges (Canny, 1986). For image editing, we propose to selectively apply the Canny control to enable precise editing in targeted regions while preserving layout consistency elsewhere.

## 3 METHOD

### 3.1 PRELIMINARIES: FLUX AND CANNY CONTROLNET

Our method builds upon FLUX, a prominent open-source T2I foundation model. FLUX uses the Diffusion Transformer (DiT) architecture (Peebles & Xie, 2023) and utilizes Rectified Flow (RF) (Liu et al., 2022) to model data-to-noise transformations. It processes multi-modal inputs via a sequence of multi-stream layers (which use separate projection matrices for text and image tokens) and single-stream layers (which use shared projection matrices). The FLUX-Canny-ControlNet (XLabs AI, 2024) integrates duplicates of two multi-stream blocks from FLUX to inject structural layout guidance into it, as shown in Figure 4.

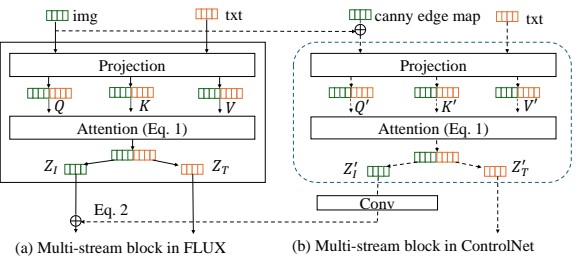

(a) Multi-stream block in FLUX     (b) Multi-stream block in ControlNet

Figure 4: The architecture of multi-stream block in FLUX and in FLUX-ControlNet.

The attention module within the FLUX multi-stream block, shown in Figure 4 (a), computes cross-attention between image tokens (I) and text tokens (T):

$$Z = \begin{bmatrix} Z_{\mathtt{I}} \\ Z_{\mathtt{T}} \end{bmatrix} = \mathrm{softmax}\left(\frac{QK^{\top}}{\sqrt{d_k}}\right)V, \quad \text{where } Q = \begin{bmatrix} Q_{\mathtt{I}} \\ Q_{\mathtt{T}} \end{bmatrix}, \; K = \begin{bmatrix} K_{\mathtt{I}} \\ K_{\mathtt{T}} \end{bmatrix}, \; V = \begin{bmatrix} V_{\mathtt{I}} \\ V_{\mathtt{T}} \end{bmatrix}. \tag{1}$$

Here, $Q, K, V$ represent the query, key, and value matrices, respectively.

Figure 4 (b) illustrates the Canny ControlNet's computational flow (dashed lines). The ControlNet is conditioned on embeddings of Canny edge map tokens and text tokens. Furthermore, the image embedding from the FLUX block is summed with the Canny edge map embedding within the ControlNet. The image token outputs from the FLUX block ($Z_{\mathtt{I}}$) and the ControlNet block ($Z'_{\mathtt{I}}$) are subsequently combined via $Z_{\mathtt{I}} \leftarrow Z_{\mathtt{I}} + \beta \cdot \mathrm{conv}(Z'_{\mathtt{I}})$, where conv denotes a $1 \times 1$ convolution layer, and $\beta$ is a hyperparameter controlling the strength of the Canny layout guidance.

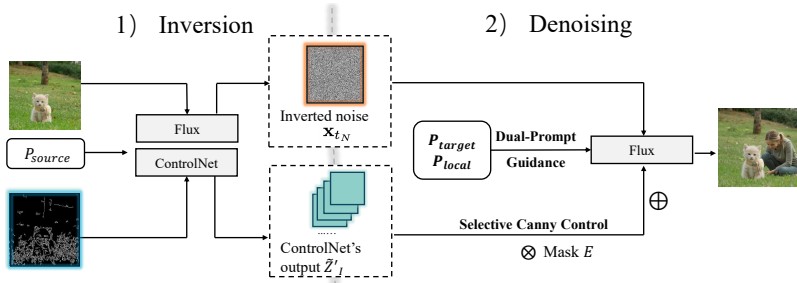

Figure 5: **The inversion-denoising process of CannyEdit.** 1) *inversion*: Starting from the source image, its Canny edge map, and a source prompt $P_{\text{source}}$, we use FireFlow to obtain the inverted noise $\mathbf{x}_{t_N}$ and corresponding Canny ControlNet outputs $\tilde{Z}'_{\mathtt{I}}$. 2) *denoising*: Using the inverted noise $\mathbf{x}_{t_N}$, we perform guided generation with selective Canny control (via a mask $E$), and dual prompts, $P_{\text{local}}$ and $P_{\text{target}}$, to provide multi-level text guidance.

### 3.2 SELECTIVE CANNY CONTROL

Image editing with diffusion models typically employs an inversion-denoising framework. This involves first inverting the source image to obtain an initial latent $\mathbf{x}_{t_N}$ that effectively captures its content at the final timestep $N$, essentially mapping the image back through the diffusion process to its noisy latent representation. We utilize FireFlow (Deng et al., 2024) for this inversion step, chosen for its favorable balance between precision and computational cost. A key aspect of our method is that, during this inversion process, while operating on the source image, we apply the Canny ControlNet and cache its outputs ($\tilde{Z}'_{\mathtt{I}}$) at relevant blocks and timesteps. These cached outputs encode structural guidance derived directly from the source image's Canny edges and features.

As illustrated in Figure 5, for the subsequent denoising phase, where the edited image is generated, we leverage these cached outputs via a technique we call *selective Canny control*. Based on a user-provided binary mask $E$ (where $E_{ij} = 1$ indicates an editable patch). We here assume that the mask $E$ is precise. Later, in Section 3.4, we introduce how CannyEdit refines imprecise masks $\hat{E}$. With $E$, we apply the cached ControlNet guidance $\tilde{Z}'_{\mathtt{I}}$ *only* to the masked region $(1 - E)$. This selective application is mathematically implemented by masking the cached ControlNet output before adding it to the output features:

$$Z_{\mathtt{I}} \leftarrow Z_{\mathtt{I}} + \beta \cdot (1 - E) \odot \text{conv}(\tilde{Z}'_{\mathtt{I}}). \tag{2}$$

By applying structural guidance from the source image's Canny edges exclusively to the unedited background region, we effectively preserve its original layout and visual details. Concurrently, the deliberate absence of Canny control within the masked editing region allows the diffusion process there to be guided primarily by the target text prompt, facilitating the desired modifications. Using the pre-computed cached outputs also enhances computational efficiency during the generation phase.

### 3.3 DUAL-PROMPT GUIDANCE

For text-aligned and seamless edits, we introduce a dual-prompt guidance strategy that modifies the attention computation to fuse two signals: a local prompt for region-specific accuracy and a global target prompt for overall contextual consistency. Although termed "dual-prompt", this strategy can accommodate multiple local prompts for simultaneous edits in several regions. For simplicity, we describe an implementation with two local prompts guiding two distinct regions; extension to more regions and prompts is straightforward.

#### 3.3.1 GENERAL FORMULATION

Let the two image regions be $\mathtt{I}_1$ and $\mathtt{I}_2$, their corresponding local prompts be $\mathtt{T}_1$ and $\mathtt{T}_2$, and the global prompt be $\mathtt{T}_*$. The number of tokens of these regions/prompts is denoted by $|\cdot|$. At each timestep, the query matrix $Q$ is formed by concatenating the query of these tokens: $Q = [Q_{\mathtt{I}_1}, Q_{\mathtt{I}_2}, Q_{\mathtt{T}_1}, Q_{\mathtt{T}_2}, Q_{\mathtt{T}_*}]$. The key $K$ and value $V$ matrices are constructed analogously.

Following Chen et al. (2024), we implement multi-level and multi-region text guidance in a training-free manner by applying regional text control via an attention mask $M$ within the self-attention

module of the FLUX blocks. The attention computation in Equation 1 becomes:

$$Z = \text{softmax}\left(\frac{QK^\top}{\sqrt{d_k}} \odot M\right)V, \quad \text{where } M = \begin{bmatrix} M_{\texttt{I}\to\texttt{I}} & M_{\texttt{I}\to\texttt{T}} \\ M_{\texttt{T}\to\texttt{I}} & M_{\texttt{T}\to\texttt{T}} \end{bmatrix} \tag{3}$$

is the attention mask applied to the concatenated image-text tokens.

For text-to-text (T2T) attention, $M_{\texttt{T}\to\texttt{T}}$, we prevent information leakage between distinct guidance signals by restricting each text prompt to attend only to its own tokens. This is enforced using a block-diagonal attention mask: $M_{\texttt{T}\to\texttt{T}} = \text{diag}(\mathbf{1}_{|\texttt{T}_1|\times|\texttt{T}_1|}, \mathbf{1}_{|\texttt{T}_2|\times|\texttt{T}_2|}, \mathbf{1}_{|\texttt{T}_\star|\times|\texttt{T}_\star|})$, where $\mathbf{1}_{n\times n}$ is an all-ones matrix of size $n \times n$. This structure enables full self-attention within each prompt while zeroing out off-diagonal blocks to disable cross-prompt interactions.

For text-to-image (T2I) attention, $M_{\texttt{T}\to\texttt{I}}$, local prompts provide guidance to their respective image regions, while the global target prompt interacts with all image regions to capture broader contextual information. This is achieved with:

$$M_{\texttt{T}\to\texttt{I}} = \begin{bmatrix} \mathbf{1}_{|\texttt{T}_1|\times|\texttt{I}_1|} & \mathbf{0}_{|\texttt{T}_1|\times|\texttt{I}_2|} \\ \mathbf{0}_{|\texttt{T}_2|\times|\texttt{I}_1|} & \mathbf{1}_{|\texttt{T}_2|\times|\texttt{I}_2|} \\ \mathbf{1}_{|\texttt{T}_\star|\times|\texttt{I}_1|} & \mathbf{1}_{|\texttt{T}_\star|\times|\texttt{I}_2|} \end{bmatrix}, \tag{4}$$

where $\mathbf{0}_{n\times m}$ denotes an all-zeros matrix. Symmetrically, we let the image-to-text (I2T) attention $M_{\texttt{I}\to\texttt{T}} = (M_{\texttt{T}\to\texttt{I}})^\top$ to allow bidirectional information flow between modalities.

Finally, for image-to-image (I2I) attention, $M_{\texttt{I}\to\texttt{I}}$, by default we only allow attention only within each respective region: $M_{\texttt{I}\to\texttt{I}} = \text{diag}(\mathbf{1}_{|\texttt{I}_1|\times|\texttt{I}_1|}, \mathbf{1}_{|\texttt{I}_2|\times|\texttt{I}_2|})$, thereby maintaining the integrity of region-specific processing.

### 3.3.2 ADJUSTMENTS FOR PRACTICAL EDITING TASKS

The default block-diagonal $M_{\texttt{I}\to\texttt{I}}$ is suitable when all regions are considered independently editable. However, for practical editing tasks, this mask can be adjusted based on the roles of different regions. For instance, when adding new content, let $\texttt{I}_1$ be the edit region and $\texttt{I}_2$ be the background region. The local prompt $\texttt{T}_2$ for the background region can be the source image's original prompt. In this scenario, $M_{\texttt{I}\to\texttt{I}}$ can be modified to:

$$M_{\texttt{I}\to\texttt{I}}(\texttt{I}_1 \to \texttt{I}_2) = \begin{bmatrix} \mathbf{1}_{|\texttt{I}_1|\times|\texttt{I}_1|} & \mathbf{1}_{|\texttt{I}_1|\times|\texttt{I}_2|} \\ \mathbf{0}_{|\texttt{I}_2|\times|\texttt{I}_1|} & \mathbf{1}_{|\texttt{I}_2|\times|\texttt{I}_2|} \end{bmatrix}. \tag{5}$$

This adjustment, through the $\mathbf{1}_{|\texttt{I}_1|\times|\texttt{I}_2|}$ block, allows the edit region $\texttt{I}_1$ to attend to the background region $\texttt{I}_2$. This enables the model to integrate contextual information from the background, leading to a more context-aware edit. This masking approach can be extended to scenarios with multiple editable regions while preserving background integrity.

To further enhance seamless blending between edited and background regions, we refine I2I attention involving the background area adjacent to the edit boundary. Let $\texttt{I}_1$ be the edit region, $\texttt{I}_:$ be the portion of the background region at the boundary of $\texttt{I}_1$, and $\texttt{I}_2$ be the remaining background region (distinct from $\texttt{I}_:$ and $\texttt{I}_1$). The I2I attention mask is then defined as

$$M_{\texttt{I}\to\texttt{I}}(\texttt{I}_1 \to \texttt{I}_2;\ \texttt{I}_1 \leftrightarrow \texttt{I}_: \leftrightarrow \texttt{I}_2) = \begin{bmatrix} \mathbf{1}_{|\texttt{I}_1|\times|\texttt{I}_1|} & \mathbf{1}_{|\texttt{I}_1|\times|\texttt{I}_:|} & \mathbf{1}_{|\texttt{I}_1|\times|\texttt{I}_2|} \\ \boxed{\mathbf{1}_{|\texttt{I}_:|\times|\texttt{I}_1|}} & \mathbf{1}_{|\texttt{I}_:|\times|\texttt{I}_:|} & \mathbf{1}_{|\texttt{I}_:|\times|\texttt{I}_2|} \\ \mathbf{0}_{|\texttt{I}_2|\times|\texttt{I}_1|} & \mathbf{1}_{|\texttt{I}_2|\times|\texttt{I}_:|} & \mathbf{1}_{|\texttt{I}_2|\times|\texttt{I}_2|} \end{bmatrix}. \tag{6}$$

The key change from Equation (5) is the introduction of the boxed block $\mathbf{1}_{|\texttt{I}_:|\times|\texttt{I}_1|}$, which allows the background region at the boundary ($\texttt{I}_:$) to attend to the edit region ($\texttt{I}_1$), enabling the model to incorporate contextual cues from the edited content into these boundary areas. Consequently, the boundary regions can better align visually and semantically with the edits, reducing artifacts and improving overall image coherence.

### 3.3.3 PROMPTING STRATEGIES FOR VARIOUS EDITING TASKS

The specifics of local prompts vary depending on the editing task. For object insertion and replacement, local prompts describe the objects to be introduced or the new objects that will substitute existing ones. For object removal, the default local prompt that we use is "empty background". We also employ classifier-free guidance (Ho & Salimans, 2022) by using descriptions of the objects targeted for removal as negative local prompts.

## 3.4 PROGRESSIVE MASK REFINEMENT FOR OBJECT ADDITION

In practice, users often lack a fine-grained binary mask $E$ for *object addition tasks* as the objects do not yet exist in the image. To improve usability, our method starts with an approximate mask $\hat{E}$, which can be an imprecise binary shape (e.g., an oval indicating the target location) or a *soft mask* derived from a user-provided point $C$, where $\hat{E}_{ij} = 1 - \|(i, j) - C\|_2$. Here, $\|(i, j) - C\|_2$ denotes the Euclidean distance between the point $(i, j)$ and $C$.

Starting from the approximate mask $\hat{E}$, We employ a two-stage process. Initially, CannyEdit operates with $\hat{E}$ until a refinement timestep, $t_{\text{refine}}$. During this first stage, the approximate mask guides the edit in Canny control and attention computation. The selective Canny control now becomes: $Z_{\text{I}} \leftarrow Z_{\text{I}} + \beta \cdot (1 - \hat{E}) \odot \text{conv}(\tilde{Z}'_{\text{I}})$. This equation grants more structural freedom near the user's hint to allow editability while maintaining stricter control elsewhere. Additionally, we incorporate $\hat{E}$ into the attention between image tokens (I) and local editing prompt tokens ($\text{T}_1$): $\frac{Q_{\text{I}}[i,j]^{\top} K_{\text{T}_1}}{\sqrt{d_k}} \leftarrow \frac{Q_{\text{I}}[i,j]^{\top} K_{\text{T}_1}}{\sqrt{d_k}} + \log(\hat{E}_{ij} + \epsilon)$, ensuring that regions near the desired edit region receive a stronger influence from the local editing prompt (a small $\epsilon$ is added to avoid $log(0)$). A similar masking strategy is applied to intra-image tokens to boost editability near the target area.

At the timestep $t_{\text{refine}}$, a refined mask, $E$, is automatically generated by applying the SAM-2 segmentation model (Ravi et al., 2024) to the denoised image. This process uses point prompts derived from the most salient locations in the aggregated cross-attention maps between image tokens and the tokens of the local editing prompt. The strategy of using segmentation models for mask refinement, also explored by Li et al. (2024b) and Tewel et al. (2025), yields a high-quality mask. Following mask refinement, the editing process continues with the precise mask, $E$, leveraging the standard selective Canny control and dual-prompt guidance as introduced previously.

This editing with mask refinement offers significant flexibility, accepting single-point hints as location indicators for where to edit. This design is key to enabling seamless, training-free integration with VLMs, which can supply the VLM-inferred point hints for edits and text inputs for complex instruction-based editing, as will be demonstrated in Section 4.

## 4 EXPERIMENT

**Experimental Setup.** We evaluate CannyEdit against state-of-the-art methods in two settings.

*Mask-Based Editing:* We compare CannyEdit with established mask-based methods to evaluate its performance in traditional region-based editing tasks including object addition, replacement and removal. Competitors include the inversion-based KV-Edit and inpainting models like BrushEdit (Li et al., 2024d), FLUX Fill (Black-Forest Labs, 2024), and PowerPaint-FLUX (Zhuang et al., 2024). For a fairer comparison, we re-trained PowerPaint on FLUX with their provided training dataset, as the original is UNet-based.

*Instruction-Based Editing:* We benchmark CannyEdit against leading open-source instruction-based editors: Step1X-Edit (Liu et al., 2025), OmniGen2 (Wu et al., 2025), BAGEL (Deng et al., 2025), FLUX.1 Kontext [dev] (Kontext) (Labs et al., 2025) , and Qwen-image-edit (Qwen-edit) (Team, 2025b). We focus on the single and multiple object addition tasks in this setup as object addition is generally more challenging. Point hints for CannyEdit are generated by GLM-4.5V (Team, 2025a) [1]. To ensure a fair comparison, we evaluate competing methods both with and without these point hints.

**Datasets.** We introduce the Real Image Complex Editing Benchmark (RICE-Bench) to address the lack of complex object interactions in existing benchmarks (Sheynin et al., 2024; Gu et al., 2024). RICE-Bench contains 80 images with challenging real-world scenarios for object addition, replacement, and removal. In addition to single-object addition (RICE-Bench-Add), we introduce RICE-Bench-Add2, a subset with 40 examples focused on the more difficult task of adding two objects in a single pass. Example images from the benchmark are shown in Figures 1, 3,7, 8 and 10, with details of data curation provided in Appendix E.

**Metrics.** We evaluate edits on three criteria: **Context Fidelity (CF)**, **Text Adherence (TA)**, and **Perceptual Realism (PR)**. CF measures background preservation using the cosine similarity between

---

[1]Detailed prompts to GLM-4.5V are provided in Appendix B.1.

Table 1: Quantitative results **under the mask-based setting** on RICE-Bench. All the methods use the same masks. The metrics are context fidelity (CF), Text Adherence (TA), and Perceptual Realism (PR). **Bold** and underlined values represent the best and second-best scores.

| Metrics | Add | | | Removal | | | Replace | | |
|---|---|---|---|---|---|---|---|---|---|
| | $CF^{\uparrow}_{\times 10^2}$ | $TA^{\uparrow}_{\times 10^2}$ | $PR^{\uparrow}_{\times 10^2}$ | $CF^{\uparrow}_{\times 10^2}$ | $TA^{\uparrow}_{\times 10^2}$ | $PR^{\uparrow}_{\times 10^2}$ | $CF^{\uparrow}_{\times 10^2}$ | $TA^{\uparrow}_{\times 10^2}$ | $PR^{\uparrow}_{\times 10^2}$ |
| KV-Edit | **93.91** | 17.25 | 65.61 | **69.81** | 16.62 | 70.12 | **64.72** | 12.36 | 65.48 |
| BrushEdit | 87.26 | 18.98 | 78.09 | 63.43 | 31.29 | 73.58 | 59.11 | 7.40 | 81.10 |
| FLUX Fill | 88.21 | 21.62 | 66.03 | 70.94 | 10.91 | 68.01 | 60.20 | 8.13 | 67.52 |
| PowerPaint-FLUX | 84.63 | 24.34 | 84.88 | 62.31 | 21.40 | 78.21 | 60.75 | 8.92 | 86.78 |
| **CannyEdit (Ours)** | 88.72 | **28.12** | **96.41** | 63.28 | **34.22** | **92.12** | 64.43 | **16.77** | **95.32** |

Table 2: Quantitative comparison on the RICE-Bench-Add (*Add*) and RICE-Bench-Add2 (*Add2*) datasets **under the instruction-based setting**, evaluating Context Fidelity (CF), Text Adherence (TA), and Perceptual Realism (PR). CannyEdit's results are based on VLM-generated point hints, differing from its mask-based results in Table 1. For fairness, competing editors were tested with and without these same hints (results with hints are marked by ⋆). Methods with CF scores below 0.70, indicating significant context failure and therefore considered ineffective edits, are marked in gray (see visual results of OmniGen2⋆ in Figures 10 for reference). **Bold** and underlined values represent the best and second-best scores among effective methods.

| Metrics | Add | | | Add2 | | |
|---|---|---|---|---|---|---|
| | $CF^{\uparrow}_{\times 10^2}$ | $TA^{\uparrow}_{\times 10^2}$ | $PR^{\uparrow}_{\times 10^2}$ | $CF^{\uparrow}_{\times 10^2}$ | $TA\ (avg.)^{\uparrow}_{\times 10^2}$ | $PR^{\uparrow}_{\times 10^2}$ |
| Step1X-Edit | 79.30 | 21.35 | 63.34 | 77.56 | 24.15 | 61.12 |
| OmniGen2 | 76.52 | 19.02 | 75.01 | 65.02 | 27.45 | 74.01 |
| BAGEL | 75.60 | 20.76 | 70.33 | 64.35 | 24.74 | 72.12 |
| Qwen-image-edit | 79.40 | 21.79 | 93.71 | 68.72 | 26.31 | 94.89 |
| FLUX.1 Kontext | 83.13 | 24.28 | **95.12** | 78.94 | 23.81 | 92.39 |
| Step1X-Edit⋆ | 81.25 | 18.33 | 61.22 | 80.75 | 22.61 | 60.59 |
| OmniGen2⋆ | 68.88 | 18.81 | 73.39 | 57.83 | 28.33 | 74.22 |
| BAGEL⋆ | 74.94 | 21.09 | 69.54 | 72.39 | 23.34 | 65.42 |
| Qwen-image-edit⋆ | 81.27 | 24.67 | 93.23 | 74.20 | 24.40 | **93.01** |
| FLUX.1 Kontext⋆ | 84.92 | 26.91 | 93.43 | 81.06 | 23.44 | 91.51 |
| **CannyEdit (Ours)** | **87.53** | **28.49** | 94.56 | **83.76** | **25.34** | 92.56 |

DINO embeddings (Caron et al., 2021) of the original and edited images. TA assesses prompt adherence via the change in GroundingDINO (Liu et al., 2024) detection confidence. For addition/replacement, this is $p_{\text{gdino}}(\text{edited image, edited object}) - p_{\text{gdino}}(\text{source image, edited object})$; for removal, the terms are reversed to measure successful elimination. PR evaluates how well the edited image align with real-world characteristics using GPT-4o (OpenAI, 2025a), which rates the visual convincingness of edits on a three-point scale (0, 0.5, 1). The prompt methodology, detailed in Appendix D.2, is adapted from Zhang et al. (2023c).

**Implementation details.** Our method is built on FLUX.1-[dev] and FLUX-Canny-ControlNet. We use 50 denoising steps, a guidance scale of 4.0, and a Canny strength $\beta$ of 0.8 for non-masked regions. Competing methods use their official default settings. Further details on hyperparameters and computational costs are provided in Appendix D.1.

## 4.1 RESULTS UNDER MASK-BASED SETTING

**Quantitative Results.** Table 1 confirms CannyEdit's superior performance in controllable local editing. Across addition, removal, and replacement tasks, CannyEdit consistently leads in both text adherence and perceptual realism while maintaining competitive context fidelity. Unlike methods such as KV-Edit, which preserve backgrounds at the great expense of perceptual realism, CannyEdit achieves an exceptional balance among all three metrics. Additional evaluation on PIE-Bench yields similar conclusion (see Appendix A.3).

**Qualitative Results.** Complementing the quantitative results with visual comparisons, Figures 7 and 8 showcase examples of CannyEdit compared against KV-Edit and the best-performing training-based method, PowerPaint-FLUX. Besides, we conducted a user study to evaluate the seamlessness of edits.

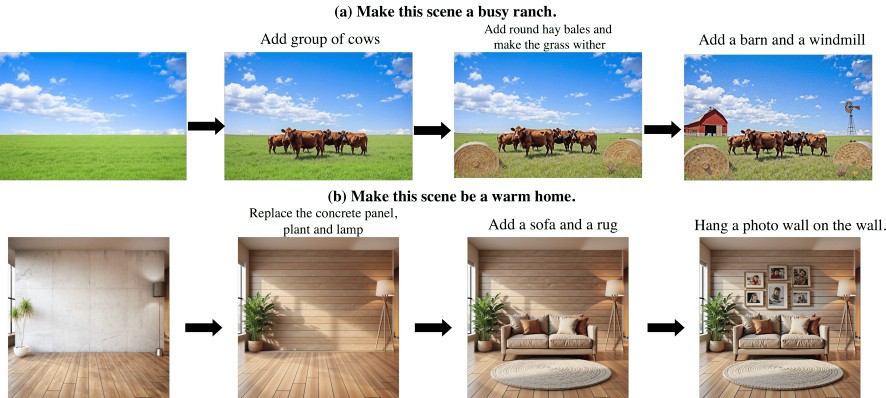

Figure 6: Chain-of-thought editing process where a VLM reasons step-by-step about edits and CannyEdit executes them. Detailed reasoning steps for this are provided in Appendix B.5.

The study revealed that CannyEdit is much less likely to be identified as AI-edited compared to other region-based methods. Further details can be found in Appendix A.2.

### 4.2 RESULTS UNDER INSTRUCTION-BASED SETTINGS

**Quantitative Results.** As shown in Table 2, our *training-free* method, CannyEdit, outperforms leading *training-based* instruction editors on both the RICE-Bench-Add (single edit) and RICE-Bench-Add2 (double edit) benchmarks. CannyEdit achieves the highest scores in context fidelity and text adherence while maintaining comparable realism. This advantage holds when competitors are given the same VLM-generated point hints, which we provided as text coordinates in the prompt—a strategy proven more effective than visual markers (see details in Appendix B.3).

**Qualitative Results.** Visualizations comparing CannyEdit with instruction-based editors using identical point hints are shown in Figure 10. Similar to the ones in Figures 1 (a,b), CannyEdit *consistently* maintains good context fidelity and text adherence across these examples, whereas the results from other methods are less consistent. CannyEdit also delivers strong perceptual realism, whereas outputs from Step1X-Edit and BAGEL exhibit noticeable degradation. Furthermore, CannyEdit enables precise, mask-based control of edit location and size (Figures 1(d), 12), a capability instruction-based methods lack even when various mask-providing strategies are attempted (Appendix B.4, Figure 13). Finally, through Chain-of-Thought (CoT) with a VLM, CannyEdit handles complex, reasoning-based edits (Figures 1(c), 6), tackling tasks that remain challenging for other editors.

### 4.3 ROBUSTNESS AND ABLATION ANALYSIS

CannyEdit maintains consistent performance regardless of visual guidance format. Comparing results on RICE-Bench-Add using user-provided masks (Table 1) versus VLM-inferred point hints (Table 2), metrics remain remarkably stable (Context Fidelity: 88.72 vs. 87.53; Text Adherence: 28.12 vs. 28.49), proving its effectiveness with both dense and sparse guidance. Appendix C presents additional component-level ablations validating our design choices and further robustness analyses.

## 5 CONCLUSION

In conclusion, this work introduces CannyEdit, a novel training-free method that addresses the core trilemma of region-based image editing—text adherence, context fidelity, and editing seamlessness. By combining selective structural control with dual-prompt guidance, CannyEdit produces high-quality edits that follow instructions while blending naturally into the original context, outperforming prior region-based methods. Its training-free design enables single-pass multi-region editing, tolerance to imprecise inputs, and seamless integration with state-of-the-art VLMs without additional training. Quantitative and qualitative results show that CannyEdit outperforms leading instruction-based editors in complex object addition tasks under controlled settings. We also outline current limitations and directions for improvement in Appendix F, pointing to promising avenues for future work.

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

# A  MORE RESULTS UNDER MASK-BASED SETTING

## A.1  VISUAL EXAMPLES

Visual examples demonstrating object addition, replacement, and removal under the mask-based setting are provided in Figure 7, which compares results generated by CannyEdit, KV-Edit, and the training-based method PowerPaint-FLUX. These examples illustrate that CannyEdit achieves a compelling balance between context fidelity, text adherence, and seamless editing quality. In contrast, KV-Edit could struggle to accurately follow the provided text prompts (as observed in (a), (b), and (f)), while outputs of PowerPaint-FLUX exhibit noticeable degradation in overall image quality. Additionally, Figure 8 provides specific examples demonstrating that, compared to CannyEdit, KV-Edit's slightly improved context fidelity comes at a significant cost to both editing seamlessness and text adherence.

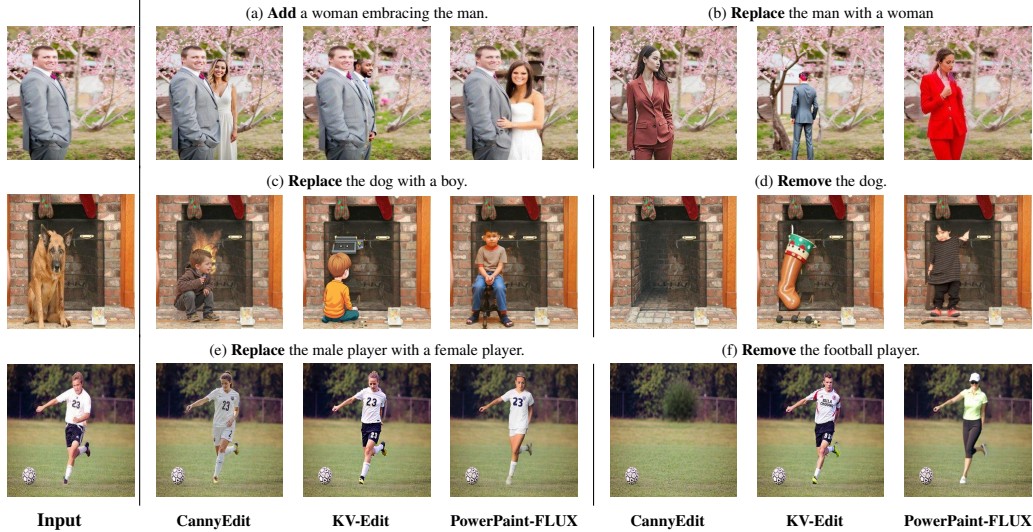

Figure 7: Visual examples of our CannyEdit, KV-Edit (Zhu et al., 2025), and PowerPaint-FLUX (Zhuang et al., 2024). across adding, removal and replacement tasks. The samples are from the RICE-Bench.

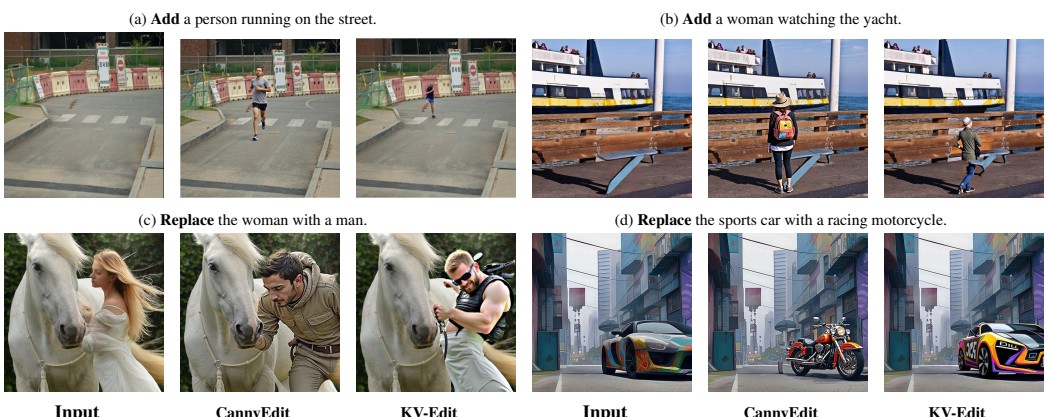

Figure 8: Visual examples demonstrating that the slightly improved context fidelity of KV-Edit (Zhu et al., 2025) comes at a significant cost to editing seamlessness and text adherence.

Table 3: Results of user study comparing CannyEdit with representative methods on RICE-Bench (± denotes 95% confidence interval). In Task1, participants identify AI-edited images from paired real (*Real*) and generated (*Gen*) images. Seamless edits yield selection ratios close to random chance (50%), as achieved by CannyEdit. In Task2, users directly compare CannyEdit against other methods; lower selection ratios for "ours" indicate superior editing seamlessness of CannyEdit.

| | General User (96 participants) | | | | Expert (41 participants) | | | |
| | Generated vs. Real | | CannyEdit vs. Others | | Generated vs. Real | | CannyEdit vs. Others | |
| Ratio regarded as AIGC (%) | Gen$^{\downarrow}$ | Real$^{\uparrow}$ | Ours$^{\downarrow}$ | Itself$^{\downarrow}$ | Gen$^{\downarrow}$ | Real$^{\uparrow}$ | Ours$^{\downarrow}$ | Itself$^{\downarrow}$ |
|---|---|---|---|---|---|---|---|---|
| KV-Edit | 86.96±7.19 | 13.04±7.19 | **37.50**±5.72 | 62.50±5.72 | 89.09±9.24 | 10.91±9.24 | **37.69**±10.2 | 62.31±10.2 |
| BrushEdit | 79.20±8.99 | 20.80±8.99 | **30.00**±4.90 | 70.00±4.90 | 82.00±12.1 | 18.00±12.1 | **19.29**±9.99 | 80.71±9.99 |
| PowerPaint-FLUX | 76.08±5.49 | 23.92±5.49 | **38.08**±6.66 | 61.92±6.66 | 88.00±6.57 | 12.00±6.57 | **33.85**±7.32 | 66.15±7.32 |
| **CannyEdit (Ours)** | **49.20**±3.56 | **50.80**±3.56 | N/A | N/A | **42.00**±8.12 | **58.00**±8.12 | N/A | N/A |

## A.2 USER STUDY

To complement the automatic metrics and systematically assess the perceived seamlessness of the edits, we conducted a user study with a total of 137 participants, comprising 96 general users with limited AIGC experience and 41 experts with formal training or experience.

The study involved two tasks where participants viewed pairs of images and were asked to identify which image was most likely AI-edited. In Task 1, an image edited by our method was compared against a real, unedited image. In Task 2, an image generated by our method is compared against one from another method. To minimize bias, the presentation order of images and questions was randomized. Only successful edits with high text adherence scores were included. A preliminary screening test filtered out participants with low accuracy in distinguishing AI-edited from real images, ensuring data reliability.

The results, presented in Table 3, highlight our method's ability to produce highly seamless edits. In Task 1, general users identified images from our method as AI-edited only 49.20% of the time, while experts did so even less frequently at 42.00%. This indicates that our edited images were often indistinguishable from real ones. Conversely, images from alternative methods were much more readily identified as AI-generated, with detection rates ranging from 76.08% to 89.09%. Consistent with Task 1, in Task 2, images generated by our method were consistently less likely to be perceived as AI-edited when compared directly to outputs from other methods, further underscoring the superior seamlessness of our editing approach.

## A.3 EXPERIMENT RESULTS ON PIE-BENCH

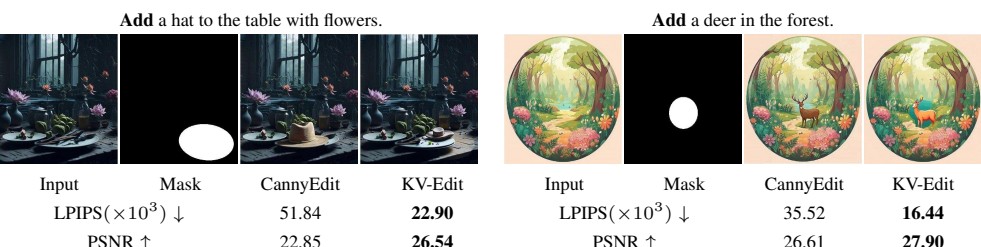

**Add** a hat to the table with flowers.  **Add** a deer in the forest.

| Input | Mask | CannyEdit | KV-Edit | Input | Mask | CannyEdit | KV-Edit |
|---|---|---|---|---|---|---|---|
| LPIPS($\times 10^3$) ↓ | | 51.84 | **22.90** | LPIPS($\times 10^3$) ↓ | | 35.52 | **16.44** |
| PSNR ↑ | | 22.85 | **26.54** | PSNR ↑ | | 26.61 | **27.90** |

Figure 9: Visual examples of our CannyEdit and KV-Edit (Zhu et al., 2025) on PIE examples along with their quantitative results on background preservation.

We extend our evaluation to PIE-Bench (Ju et al., 2024) which involves more images in various editing tasks. Following Zhu et al. (2025), we use seven metrics: HPSv2 (Wu et al., 2023) and aesthetic scores (Schuhmann et al., 2022) (image quality), PSNR (Huynh-Thu & Ghanbari, 2008), LPIPS (Zhang et al., 2018), and MSE (background preservation), and CLIP score (Radford et al., 2021) and Image Reward (Xu et al., 2023) (text adherence). Following Li et al. (2024d); Xu et al. (2024); Zhu et al. (2025), we exclude the style transfer task to focus on region-based image editing.

Table 4: Comparison with other methods on PIE-Bench. VAE$^*$ denotes the inherent reconstruction error through VAE reconstruction only. Except the result of ours, other results follow (Zhu et al., 2025).

| Metrics | Image Quality | | Background Preservation | | | Text Adherence | |
|---|---|---|---|---|---|---|---|
| | HPS$^\uparrow_{\times 10^2}$ | AS$^\uparrow$ | PSNR$^\uparrow$ | LPIPS$^\downarrow_{\times 10^3}$ | MSE$^\downarrow_{\times 10^4}$ | CLIP Sim$^\uparrow$ | IR$^\uparrow_{\times 10}$ |
| VAE$^*$ | 24.93 | 6.37 | 37.65 | 7.93 | 3.86 | 19.69 | -3.65 |
| P2P (Hertz et al., 2022) | 25.40 | 6.27 | 17.86 | 208.43 | 219.22 | 22.24 | 0.017 |
| MasaCtrl (Cao et al., 2023) | 23.46 | 5.91 | 22.20 | 105.74 | 86.15 | 20.83 | -1.66 |
| RF-Inversion (Rout et al., 2024) | **27.99** | **6.74** | 20.20 | 179.73 | 139.85 | 21.71 | 4.34 |
| RFSolver-Edit (Wang et al., 2024) | _27.60_ | _6.56_ | 24.44 | 113.20 | 56.26 | 22.08 | 5.18 |
| KV-Edit (Zhu et al., 2025) | 27.21 | 6.49 | **35.87** | **9.92** | **4.69** | 22.39 | 5.63 |
| BrushEdit (Li et al., 2024d) | 25.81 | 6.17 | 32.16 | _17.22_ | _8.46_ | _22.44_ | 3.33 |
| FLUX Fill (Black-Forest Labs, 2024) | 25.76 | 6.31 | _32.53_ | 25.59 | 8.55 | 22.40 | _5.71_ |
| **CannyEdit (Ours)** | 27.19 | 6.38 | 32.18 | 26.38 | 9.79 | **25.36** | **8.20** |

The result is displayed in Table 4. As for text alignment, CannyEdit significantly outperforms other methods both in CLIP similarity($22.44 \rightarrow$ **25.36**) and Image Reward($5.71 \rightarrow$ **8.20**). Our method also keeps a competitive level in image quality. The accuracy of masked region preservation achieved by CannyEdit is numerically inferior to that of KV-Edit. However, as illustrated by the examples in Figure 9, the background preservation metrics tend to penalize the more natural and complete results generated by CannyEdit, which may not effectively reflect the quality of the background preservation.

## B  MORE RESULTS UNDER INSTRUCTION-BASED SETTING

### B.1  PROMPTING FOR POINT HINTS OF EDITING

In experiments on RICE-Bench-Add and RICE-Bench-Add2, we prompt GLM-4.5V (Team, 2025a) to provide point hints that indicate approximate regions for editing operations. The detailed prompt are as below:

```
Let's say I want to add two new subjects to an image.

Subject A: {Description to subject A to add}.

Subject B: {Description to subject B to add}.

Could you suggest a point coordinate (x,y) for placing each of these
two subjects? The coordinate should be normalized between 0 and 1,
where 0 to 1 means left to right (x) and top to bottom (y).

First consider the position of related objects in the image.

Then, analyze where the added subjects should be positioned (left,
right, above, or below existing elements) and assign a
specific point coordinate where each subject should be centered.

The points should be in areas where the subjects wouldn't overlap
with existing elements.

Please provide the coordinates in the format:
Subject A: [x,y], Subject B: [x,y].
```

Prompt Example 1: Instruction prompt for getting the point hints of edits.

## B.2 MORE VISUAL COMPARISONS BETWEEN CANNYEDIT AND INSTRUCTION-BASED EDITORS

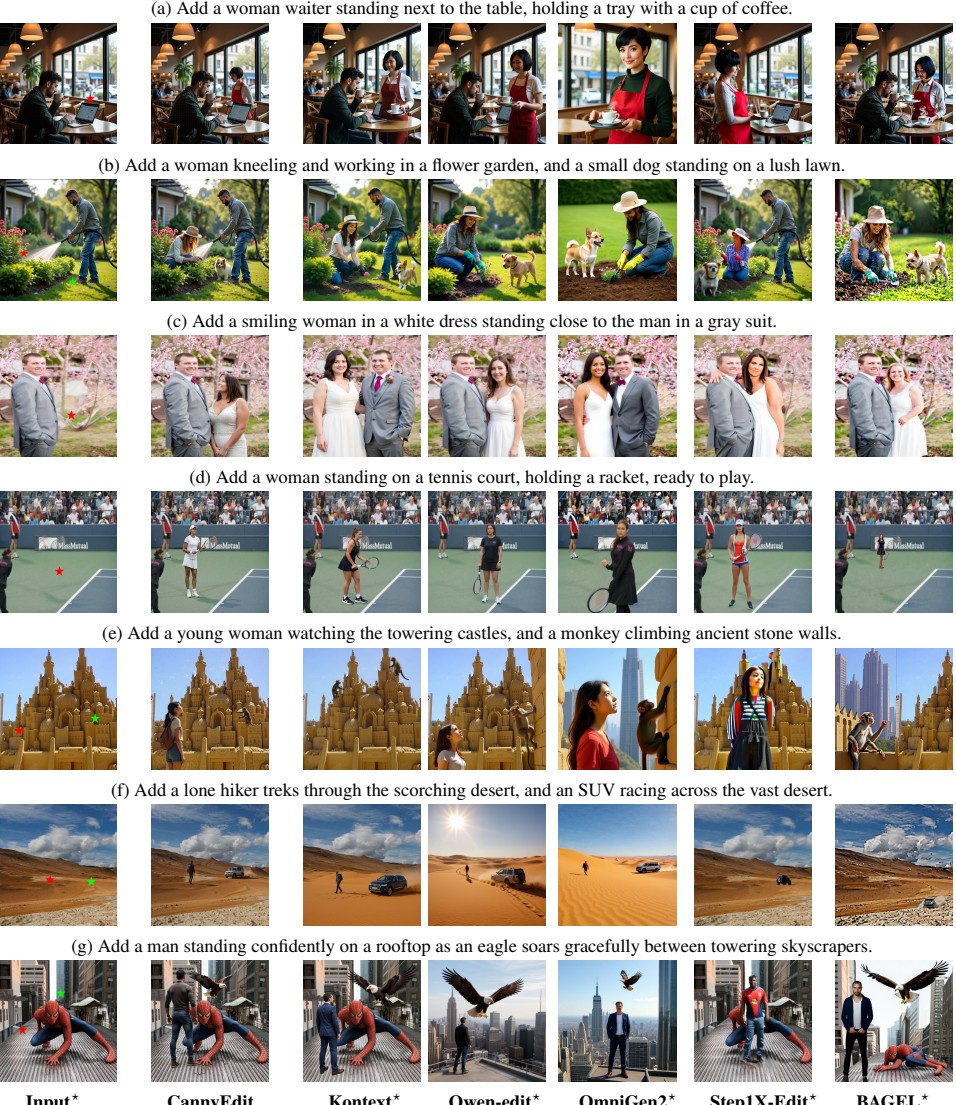

Figure 10: Visual comparison between CannyEdit and instruction-based editors using equivalent point hints (the point hints are marked as red/green stars in input*) on RICE-Bench-Add/Add2 examples. For instruction-based editors, point information is provided as text coordinates.

## B.3 EVALUATION OF DIFFERENT POINT-PROVIDING STRATEGIES FOR INSTRUCTION-BASED EDITORS

To ensure a fair comparison, we evaluated two distinct strategies for providing VLM-generated point hints to baseline instruction-based editors. The experiments on RICE-Bench-Add/Add2 were conducted using three settings: (a) the instruction prompt only; (b) the instruction prompt augmented with text coordinates from VLM-inferred hints (e.g., *"Near the point (0.6, 0.644), add an elderly man..."*); and (c) the input image marked with a visual prompt (a red star, ⋆) which is referenced in the text (e.g., *"Near the red star in the image, add an elderly man..."*). While the main paper reports results for settings (a) and (b) in Table 2, this section provides a direct comparison between the two hint-providing strategies: text coordinates (b) and visual markers (c).

Table 5: Quantitative comparison of two strategies for providing point hints to instruction-based editors on the RICE-Bench-Add benchmark. **Setting 1** embeds coordinates into the text instruction, while **Setting 2** uses visual star markers on the image. The results show that Setting 1 generally achieves better performance, particularly for the Text Adherence metric.

| Metrics | Context Fidelty$^{\uparrow}_{\times 10^2}$ | Text Adherence$^{\uparrow}_{\times 10^2}$ |
|---|---|---|
| Setting 1: Include the text coordinate into the text instruction. | | |
| Step1X-Edit* | 81.25 | 18.33 |
| OmniGen2* | 68.88 | 18.81 |
| BAGEL* | 74.94 | 21.09 |
| Qwen-image-edit* | 81.27 | 24.67 |
| FLUX.1 Kontext* | 84.92 | 26.91 |
| Setting 2: Mark the point hints as stars in the input image and mention the star in the text | | |
| Step1X-Edit* | 80.60 | 19.10 |
| OmniGen2* | 63.76 | 18.01 |
| BAGEL* | 70.29 | 15.42 |
| Qwen-image-edit* | 82.21 | 22.38 |
| FLUX.1 Kontext* | 85.12 | 25.85 |

Table 5 presents the quantitative results comparing these two approaches. The data indicates that embedding coordinates directly into the text instruction generally yields better text adherence. Also, as qualitative examples shown in Figure 11. the visual marker strategy often leaves the star artifact in the generated images. We therefore use the text coordinate strategy for all baseline comparisons in our main analysis.

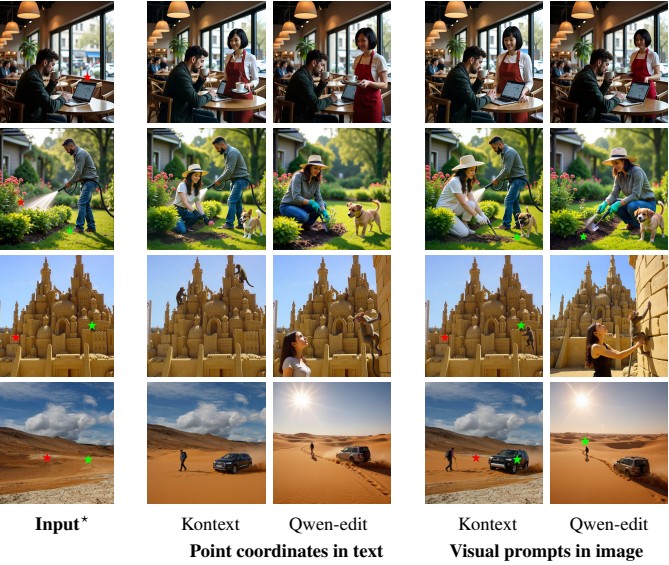

Figure 11: Qualitative comparison of two point-providing strategies for FLUX.1 Kontext and Qwen-image-edit. **Setting 1** includes point coordinates in the text instruction, while **Setting 2** uses a visual star marker in the input image. We can see that Setting 2 frequently leaves the star artifact in the generated image (rightmost 2 columns), degrading the output quality visually.

### B.4 CANNYEDIT VS. INSTRUCTION-BASED EDITORS FOR MASK-GUIDED EDITING TASKS

Figure 12 demonstrates that CannyEdit effectively supports edits at various scales when masks of corresponding sizes are provided.

In comparison, in Figure 13, we try three ways of supplying the mask information to the instruction-based editors, FLUX.1, Kontext [dev], and Qwen-image-edit, but none of them successfully follows the mask to make precise edits in the user-specified region.

To further quantitatively evaluate this, we conduct experiments on the RICE-Bench-Add examples. Masks are provided to FLUX.1, Kontext [dev], and Qwen-image-edit by marking the image with an oval that has a red border and no fill (the strategy (C) in Figure 12). This oval indicates the edit region and is used with the prompt, *"add ... within the red oval and remove the red oval."*. After the editing process, we compute the Intersection over Union (IoU) between the pixels of the generated objects (segmented by LanguageSAM (Medeiros, 2024)) and the pixels covered by the red-bordered oval. The results are as:



**CannyEdit: 0.89    Kontext: 0.68    Qwen-Edit: 0.34**



The results show that CannyEdit achieves substantially higher spatial precision in placing added objects within the specified region, outperforming both Kontext and Qwen-Edit by a wide margin. The visualizations in Figure 14 show that while CannyEdit generates objects that adhere well to the given mask region, Kontext tends to generate objects that are larger than the specified region in all three examples. In contrast, Qwen-Edit frequently misplaces the object, generating it outside the designated area.

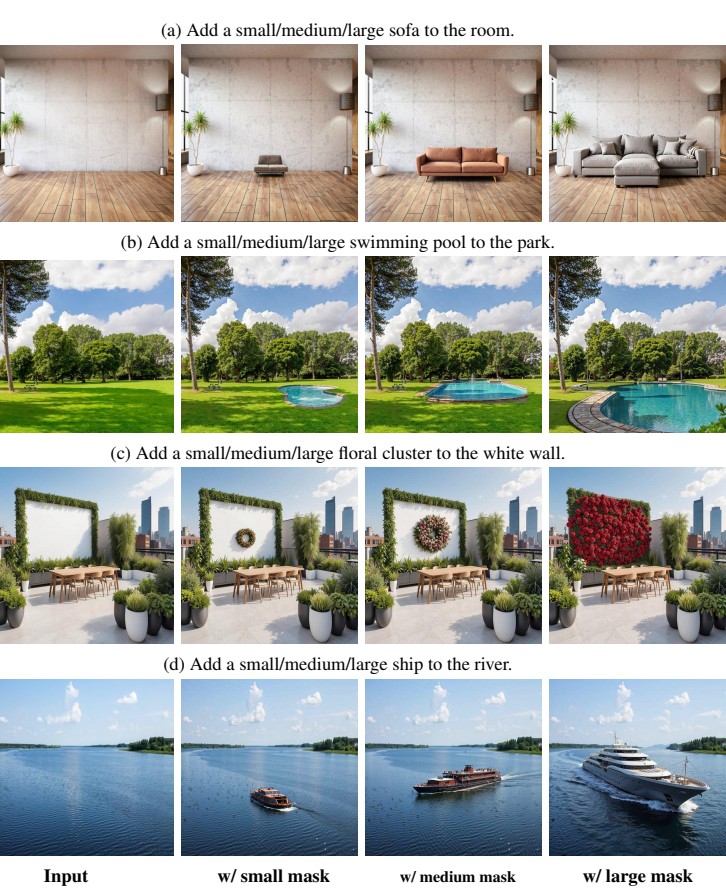

(a) Add a small/medium/large sofa to the room.

(b) Add a small/medium/large swimming pool to the park.

(c) Add a small/medium/large floral cluster to the white wall.

(d) Add a small/medium/large ship to the river.

| **Input** | **w/ small mask** | **w/ medium mask** | **w/ large mask** |

Figure 12: CannyEdit enables precise, mask-guided local editing, generating objects at specified locations and scales from masks of varying sizes (masks omitted here; examples of the masks shown in Figure 13)

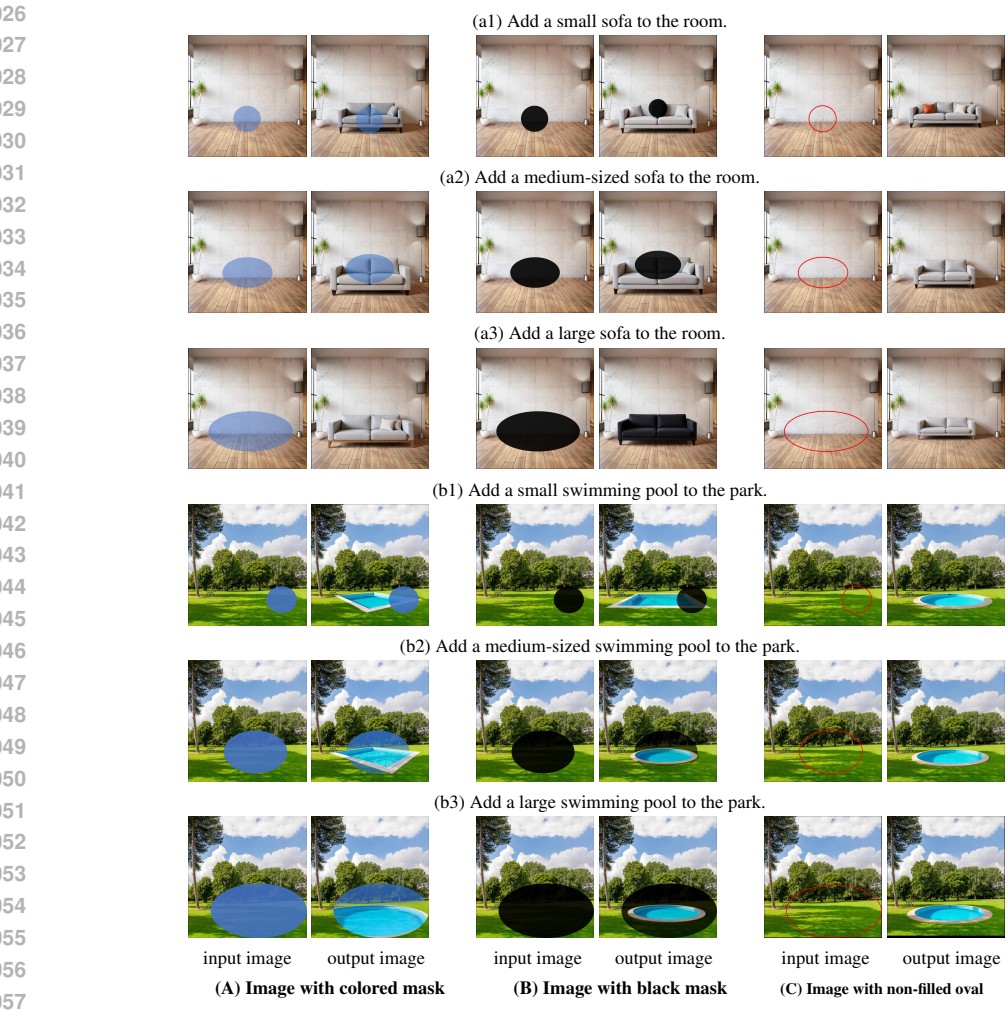

Figure 13: Analysis of strategies for providing mask information to instruction-based editors. For case (a) with sofa, we used FLUX-Kontext.1 [Dev]; for case (b) we used Qwen-image-edit. We tested three strategies: (A) colored masks with prompt "add ... within the colored mask", (B) black masks with similar prompting, and (C) ovals with a red border and no fill with prompt "add ... within the red oval and remove the red oval." None of these methods enabled the editors to precisely follow the masks to generate objects with desired sizes. Strategies (A) and (B) often preserved the mask as an artifact, while strategy (C) successfully removed the red oval but failed to generate content matching the specified size.

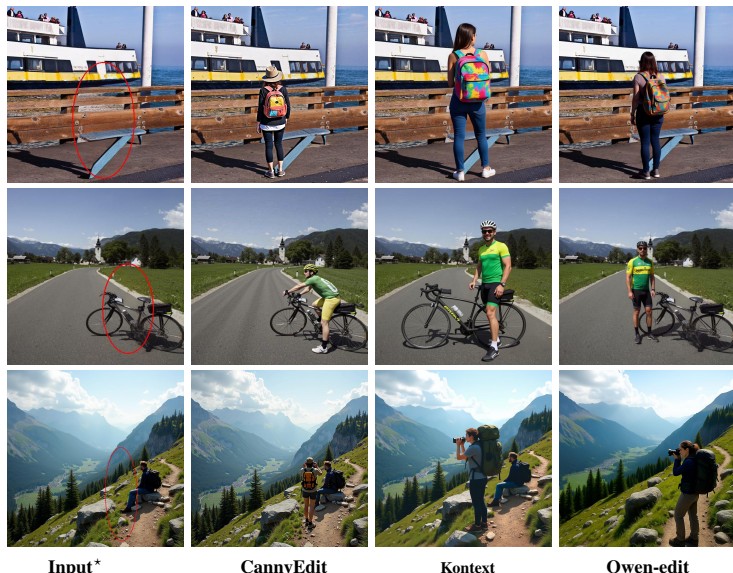

|  |  |  |  |
|---|---|---|---|
| **Input*** | **CannyEdit** | **Kontext** | **Qwen-edit** |

Figure 14: Visualizations of fitting a mask region defined by ovals with a red border and no fill, using the prompt "add ... within the red oval and remove the red oval." to the instruction-based editors. The results show that FLUX.1 Kontext consistently generate objects larger than the given region in all three examples, while Qwen-Image-Edit misplaces the object beyond the designated region in the second and third examples.

### B.5 MORE RESULTS: CANNYEDIT INTEGRATED WITH SOTA VLMs

#### B.5.1 USING VLM AS AN REASONER AND CANNYEDIT AS AN EXECUTOR

In Figures 1(c) and 6, we demonstrate how CannyEdit supports complex, reasoning-based edits through a Chain-of-Thought (CoT) process using a VLM reasoner. This process involves the VLM reasoning about what to edit and where to make the changes, guided by abstract instructions and the outcomes of each editing step. CannyEdit then performs the precise edits accordingly.

Here, we illustrate the prompting and reasoning process with GPT-5 (OpenAI, 2025b), which was tasked with interpreting the instruction, "Make this scene a busy ranch," for Figure 6(a).

**Make this scene a busy ranch.**

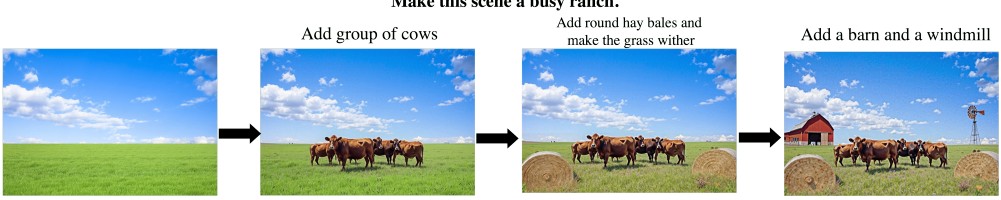

Add group of cows | Add round hay bales and make the grass wither | Add a barn and a windmill

```
User:

<source image> I want to make this scene a busy ranch.

Please help me by suggesting one edit at a time in a step-by-step manner.

For each edit you provide, please include the following:

1. Analysis: Explain why this particular edit is important for making the
      scene feel more like a busy ranch.
2. Edit Description: Clearly describe what visual element or object
      should be added, removed, or modified.
```

```
3. Location: Specify the center coordinates of the edit using normalized
    values between 0 and 1 (where (0, 0) is the top-left corner and (1,
    1) is the bottom-right corner).

After you suggest the first edit and explain it thoroughly,
I will show you the updated scene.

Then, based on the result, you can analyze what to edit next.

Let us proceed step by step.

GPT-5:

Analysis: Herd animals are the core visual cue of a working ranch.
    Placing a small group of cows in the center instantly communicates
    livestock activity and gives us a focal point to build around (
    fencing, ranch hands, troughs, etc.). Multiple cows instead of a
    single one conveys busy and sets scale for future edits.

Edit Description: Add a group of 4 to 6 grazing cows standing and grazing
    , with a few facing different directions. Include subtle shadows on
    the grass consistent with the sun coming from top-left. Keep their
    size appropriate to a mid-ground placement so they do not dominate
    the frame.

Location: Center at (0.50, 0.58) with a spread radius of ~0.12 to
    distribute the group naturally across the mid-ground.

User:

<edited image 1>  Here is the updated scene, please suggest the next edit
    .

GPT-5:

Analysis: Hay bales are classic ranch infrastructure, signaling ongoing
    feeding and harvest activity. Placing round bales near the foreground
     corners frames the herd, adds depth, and enhances the busy feel with
     signs of recent work. Slightly withering patches of grass around the
     herd and bales suggest active grazing and trampling, adding realism.

Edit Description:

Add two large round hay bales (golden-brown, rough straw texture, visible
     spiral pattern). Include short cast shadows falling slightly down-
    right to match lighting.

Left bale and right bale fully visible near the edge.

Locally desaturate and darken grass to a straw-tinged green-brown in
    trodden patches: a worn oval around the cows and smaller scuffed
    areas near each bale to imply feeding/traffic.

Location:
Left hay bale center: (0.12, 0.86), diameter ~0.15 of image width.
Right hay bale center: (0.88, 0.88), diameter ~0.15 of image width.
Worn grass around herd: oval centered at (0.50, 0.62), radius ~0.22*0.12.

User:

<edited image 2>  Here is the updated scene, please suggest the next edit
    .

GPT-5:
```

```
Analysis: Adding core ranch structures anchors the scene as a working
    operation rather than a random pasture. A classic red barn signals
    storage, shelter, and daily activity. A metal windmill provides
    utility and vertical interest, common on cattle ranches for filling
    stock tanks. Placing them on opposite sides of the horizon balances
    the composition and builds depth without distracting from the herd.

Edit Description:

Barn: Add a medium-distant, traditional gabled barn with red siding, and
    a sliding door partially open. Keep scale consistent with being
    several hundred meters away; soft edges due to distance haze.
Windmill: Add a tall, slender lattice-tower windmill with a multi-blade
    wheel and tail vane.  Base should meet the ground near the horizon

Location:

Barn center: (0.18, 0.28). Width ~0.16 of image width; base aligned just
    above the horizon line.
Windmill center: (0.82, 0.20). Tower height ~0.22 of image height; base
    at (0.84, 0.36).
```

Prompt Example 2: Prompting and reasoning process of GPT-5 for Fiugre 6 (a).

### B.5.2 USING VLM AS A LOCATOR AND CANNYEDIT AS AN EXECUTOR

Replacement and removal tasks that require precise grounding are challenging for instruction-based editors, especially in complex scenes. In Figure 15, we demonstrate that by using GLM-4.5V (Team, 2025a) as a locator, CannyEdit, acting as an executor, effectively solves cases where FLUX.1 Kontext [dev] and Qwen-Image-Edit fail.

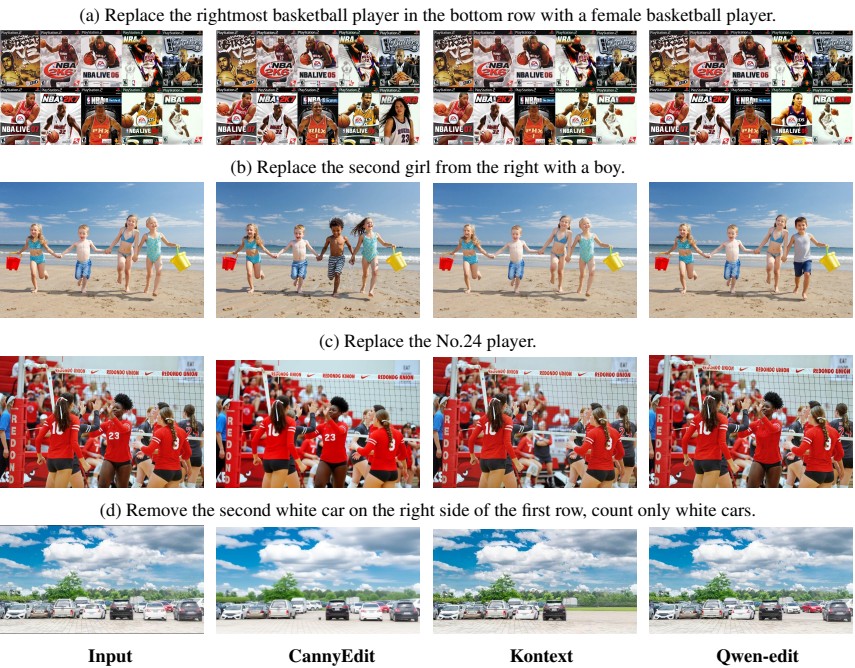

Figure 15: Visual examples illustrating the power of CannyEdit as a precise image editing executor. Guided by a sophisticated locator (GLM-4.5V), CannyEdit successfully performs complex removal and replacement tasks where FLUX.1 Kontext and Qwen-image-edit fail.

# C ABLATION STUDY

## C.1 ROBUSTNESS OF CANNYEDIT TO INPUT VISUAL HINTS

In region-based editing, we use oval masks in RICE-Bench-Add to indicate the target edit region. Below, we report Context Fidelity (CF) and Text Adherence (TA) across different mask configurations: the default oval masks, simple rectangular masks, and oval/rectangular masks augmented with random boundary perturbations to emulate imprecise, hand-drawn inputs. We consider two augmentation regimes: (1) high-frequency, small variations (fine brush strokes) and (2) low-frequency, larger variations (broad brush strokes).

Beyond masks, we also evaluate robustness to sparse spatial hints by comparing user-provided oval masks (Table 1) against VLM-inferred point hints (Table 2) on RICE-Bench-Add. The metrics remain highly stable between the two settings (CF: 88.72 vs. 87.53; TA: 28.12 vs. 28.49), demonstrating effectiveness under both dense and sparse guidance. Results are summarized below:

| Input visual hints | Augmentation | Context Fidelity (CF) | Text Adherence (TA) |
|---|---|---|---|
| Oval Mask | / | 88.72 | 28.12 |
| | High-freq, smaller variations | 87.69 | 27.98 |
| | Low-freq, larger variations | 86.21 | 28.29 |
| Rectangular Mask | / | 86.22 | 28.23 |
| | High-freq, smaller variations | 85.95 | 28.36 |
| | Low-freq, larger variations | 85.34 | 28.52 |
| VLM-inferred Point | / | 87.53 | 28.49 |

We further quantify the importance of the mask refinement procedure described in Section 3.4 for both standard oval masks and VLM-inferred point hints:

| Input visual hints | Mask Refinement | Context Fidelity (CF) | Text Adherence (TA) |
|---|---|---|---|
| Oval Mask | Yes | 88.72 | 28.12 |
| | No | 84.12 | 29.31 |
| VLM-inferred Point | Yes | 87.53 | 28.49 |
| | No | 70.22 | 29.50 |

These results underscore the effectiveness and necessity of mask refinement for preserving image context, especially when only sparse point hints are provided.

## C.2 ABLATIONS ON CANNY CONTROL AND DUAL-PROMPT GUIDANCE

We ablate key components of CannyEdit using addition and replacement tasks from RICE-Bench under user-provided oval masks. The results are shown in the table below (*CC* denotes Canny Control and *DP* denotes Dual-Prompt guidance). For selective Canny control, we test: (i) using selective Canny ControlNet outputs computed from the current T2I denoising process instead of using the cached ControlNet outputs, (ii) removing Canny control entirely, and (iii) using full cached Canny ControlNet outputs without selectivity. The first two variants substantially reduce context fidelity, while the third improves context preservation but weakens text adherence. Ablations on dual-prompt guidance show that both local and global prompts are necessary for balanced performance.

| | | Add | | Replace | |
|---|---|---|---|---|---|
| | | $\mathrm{CF}^{\uparrow}_{\times 10^2}$ | $\mathrm{TA}^{\uparrow}_{\times 10^2}$ | $\mathrm{CF}^{\uparrow}_{\times 10^2}$ | $\mathrm{TA}^{\uparrow}_{\times 10^2}$ |
| **CannyEdit (Ours)** | | 88.72 | 28.12 | 64.43 | 16.77 |
| Variants of *CC* | selective *CC* from current T2I | 84.81 | 27.93 | 55.24 | 13.89 |
| | w/o *CC* | 79.73 | 26.29 | 51.51 | 11.83 |
| | full *CC* | 93.41 | 19.73 | 64.02 | 12.51 |
| Variants of *DP* | local prompt only | 93.43 | 10.23 | 58.92 | 17.09 |
| | target prompt only | 85.04 | 24.96 | 60.82 | 15.43 |

## C.3 CHOICE OF CONTROLNET MODALITY

Our method builds upon the Canny ControlNet. We choose this modality to serve the primary objectives of region-based editing: preserve the untouched context as faithfully as possible while enabling strong generative flexibility within the edited region. Canny maps capture the core structural cues of the source image—edges and contours—which, when paired with our inversion strategy, effectively maintain the original composition and fine details. By selectively disabling Canny guidance inside the edit mask, the model gains the necessary freedom to follow the text prompt without compromising the surrounding content.

To support this choice, we compare the available FLUX-ControlNet modalities (Canny, HED, Depth) and observe that Canny provides the most favorable balance between context preservation and instruction following. Although Depth yields marginally higher text adherence, it incurs a substantial loss in context fidelity, rendering it less suitable for high-fidelity edits. The experiment is conducted on RICE-Bench-Add using user-provided masks.

| Modality | Context Fidelity (CF) | Text Adherence (TA) |
|---|---|---|
| Depth | 83.37 | 29.01 |
| HED | 81.32 | 28.33 |
| Canny | 88.72 | 28.12 |

## C.4 HYPERPARAMETER STUDIES

The hyperparameter studies are conducted on RICE-Bench-Add using user-provided masks.

**ControlNet strength** ($\beta$). We use $\beta = 0.8$ by default (as in the original Canny ControlNet). Decreasing the strength mildly improves text adherence but reduces context fidelity. The default setting provides the best balance for high-quality, context-preserving edits.

| ControlNet Strength ($\beta$) | Context Fidelity (CF) | Text Adherence (TA) |
|---|---|---|
| 0.3 | 84.15 | 29.42 |
| 0.5 | 85.87 | 29.59 |
| 0.8 (default) | 88.72 | 28.12 |

**Number of denoising steps**. Here we evaluate the trade-off between computational cost and performance. Our results show that even when reducing the workload to 30 or 40 steps, our model's Text Adherence (25.98 and 26.62, respectively) remains significantly higher than both the KV-Edit (17.25) and PowerPaint-FLUX (24.34) baselines. A more significant performance drop is only observed at 20 steps. Therefore, while 50 steps yield optimal quality, we recommend a minimum of 30 denoising steps for a robust balance between high-quality output and computational efficiency.

| # Steps | Context Fidelity (CF) | Text Adherence (TA) |
|---|---|---|
| 20 | 90.12 | 23.58 |
| 30 | 89.12 | 25.98 |
| 40 | 88.51 | 26.62 |
| 50 (default) | 88.72 | 28.12 |
| KV-Edit (baseline) | 93.91 | 17.25 |
| PowerPaint-FLUX (baseline) | 84.63 | 24.34 |

# D IMPLEMENTATION DETAILS

## D.1 METHOD EXECUTION DETAILS

We implement our method based on FLUX.1-[dev] (Black-Forest Labs, 2024) and FLUX-Canny-ControlNet (XLabs AI, 2024), using 50 denoising steps and a guidance value of 4.0. The strength parameter of Canny control $\beta$ is set to 0.8 in the inversion and for the non-mask-boundary background region. Other methods were implemented based on their official code releases' default settings unless otherwise specified.

All evaluation experiments were conducted on a machine with NVIDIA A100 GPUs and an AMD EPYC 7642 Processor (with a total of 96 cores). Our CannyEdit, using FLUX.1-[dev] and FLUX-Canny-ControlNet, can be implemented on a single NVIDIA A100 GPU (or other GPUs with approximately 40 GB of memory for a single editing task). Multi-editing tasks in one generation pass require more memory due to the involvement of additional text tokens.

Compared to previous training-free image editing methods based on FLUX (Wang et al., 2024; Deng et al., 2024; Tewel et al., 2025), the additional computational cost of our method primarily arises from the integration of a Canny ControlNet. However, the FLUX-Canny-ControlNet, with only 0.74B parameters, is lightweight compared to the FLUX model, which has 12B parameters. This is because the Canny ControlNet includes only two multi-stream blocks, whereas FLUX contains 19 multi-stream blocks and 38 single-stream blocks. The computation overhead introduced by our method is acceptable, as the lightweight nature of the Canny ControlNet ensures efficient performance without significantly increasing resource demands.

Compared to leading instruction-based methods, FLUX.1 Kontext [dev], which also has 12B parameters, and Qwen-Image-Edit, which has 20B parameters. Our CannyEdit approach involves inversion, which requires additional computation. As a result, its computational cost and runtime are higher than those of FLUX.1 Kontext [dev]. However, despite the additional inversion cost, our method runs significantly faster than Qwen-Image-Edit. On the machine equipped with an A100 GPU, performing a single edit on a 512×512 image with 50 denoising steps using CannyEdit takes an average of approximately 29 seconds, whereas Qwen-Image-Edit requires around 74 seconds for the same task (we use their official *diffusers* implementation).

## D.2 EVALUATION DETAILS

Perceptual Realism (PR) assesses how well an edited image aligns with real-world visual characteristics. To evaluate this, we employ GPT-4o (OpenAI, 2025a), which rates the visual plausibility of edits on a three-point scale (0, 0.5, 1). Our prompting methodology is adapted from Zhang et al. (2023c) and is detailed below.

```
You are now an evaluator responsible for assessing the Perceptual
Realism (PR) of an AI-generated image. Please assign a score from the
set: [0, 0.5, 1].

General Guidelines for Perceptual Realism (PR) Scoring:

PR = 0: The image contains obvious distortions or artifacts that make it
unrecognizable at first glance.

PR = 0.5: The image has some artifacts, but the objects are still
recognizable, or it has an unnatural sense of detail in certain areas
(i.e., the image looks strange only after close examination).

PR = 1: The image appears generally realistic. It does not need to be
100% perfect, approximately 90% realism is acceptable.

Please analyze the given image based on the above rules, and provide your
reasoning. Finally, assign a PR score from the set [0, 0.5, 1].
```
Prompt Example 3: Prompt to GPT-4o for evaluating the perceptual realism of the edited images.

# E CURATION OF RICE-BENCH

In this work, we focus on real-world image editing scenarios that involve complex interactions between the edited region and the surrounding image context. However, existing real-world image benchmarks (Sheynin et al., 2024; Gu et al., 2024) primarily involve edits to minor objects and lack realistic interactions among objects (e.g., "add a glass of water on the table"). To address this limitation, we introduce the **R**eal **I**mage **C**omplex **E**diting Benchmark (**RICE-Bench**), designed to better evaluate *context fidelity*, *text adherence*, and *editing seamlessness* in real-world editing tasks.

Compared to previous benchmark, the edits in RICE-Bench typically involve more significant changes to the image layout, posing greater challenges in balancing context fidelity and text adherence.

RICE-Bench consists of 80 images depicting real scenes with complex editing scenarios, divided into adding, replacement, and removal tasks (30 for adding, and 25 each for replacement and removal). Each example includes a source image, an input mask, a source prompt for description of original image, a local prompt for object to be edited, and a target prompt describing image after editing as in Figure 16. Besides single object-addition (RICE-Bench-Add), we manipulate RICE-Bench-Add2, a subset with another 40 examples for better evaluation of adding two objects in a single forward pass. Example images from the benchmark are shown in Figures 1, 3,7, 8 and 10. They share the same curation pipeline but each sample in RICE-Bench-Add2 has one more local prompt and corresponding mask.

In summary, we curated the datasets from three sources. Two of them are open datasets: PIPE (Wasserman et al., 2024), which contains semi-synthetic inpainting data, and PIE (Ju et al., 2024), which includes some natural images. We also utilize a mainstream LLM (DeepSeek-V3 (DeepSeek-AI et al., 2025)) and an image generation model (FLUX (Black-Forest Labs, 2024)) to construct and filter high-quality synthetic images. The LLM and a Vision-Language Model (VLM), Qwen2-VL-72B (Team, 2024), are used to create local editing prompts and target prompts. Based on these sources, the data curation pipeline can be divided into four steps:

1. **Source image construction and filtering**. For the PIPE (Wasserman et al., 2024) and PIE (Ju et al., 2024) sources, we carefully filter images with relatively complex real-life scenarios and interactions. For synthetic data generation, as referred to in (Tewel et al., 2025), we prepare an instruction prompt to ask the LLM to generate a set of text prompts, which are utilized to generate source images by FLUX. The instruction prompt is shown in *Prompt Example 4*. With synthesized source images, we further filter high-quality samples without distortion or irrational content.

2. **Mask generation**. For the adding task, we use *cv2.polylines()* and *cv2.fitEllipse()* in OpenCV (Bradski, 2000) to manually draw oval masks for possible locations to add objects. For RICE-Bench-Add2, there are two masks. Each for one object to be edited respectively. For replacement and removal tasks, we use LanguageSAM (Medeiros, 2024) to segment target objects based on their names.

3. **Local prompt generation**. For the adding task, we manually describe one or two objects to be added and ask the VLM (Qwen2-VL-72B) to enrich the details of the description. For replacement and removal tasks, as referred to in (Zhuang et al., 2024), given the source image and corresponding editing masks, we first crop the target object and then utilize the VLM to describe the object only.

4. **Source prompt and target prompt generation**. Given the source image, we first ask the VLM to caption the image as the source prompt. Based on the source prompt, mask, and corresponding local prompts, we use the VLM to generate a target prompt that describes the whole image after editing. For the adding task, an example of the instruction prompt used to generate the target prompt is shown in *Prompt Example 5*.

```
Please generate a JSON list of 100 sets. Each set consists of:
an index, a source prompt.
The source prompt describes a source image.
The source prompt should include a relatively complex real-life
scenario and at least one person.
Here is an example:
{
    "index": 1,
    "src_prompt": "A beautiful park with a bench, a man is sitting
                  on it"
}
```

Prompt Example 4: Instruction prompt for generating texts to create source images.

```
Given the caption of an image: {source_prompt}, Now I want to add
{num_objects} objects or persons.
Their corresponding descriptions are: {all_local_prompts}.
According to these caption and descriptions, please summarize them
into a refined target prompt within 20 words.
```

```
    Return the target prompt only.
```

Prompt Example 5: Instruction prompt for generating target prompts.

(a) **Add** a person running on the street.

**SourceP:** A construction site with barriers and signs.

**TargetP:** A construction site with barriers and signs, with a person jogging through the area.

**LocalP:** A person running on a path, wearing athletic shoes and shorts.

(b) **Replace** the pepper with three apples.

**SourceP:** A red pepper on a towel.

**TargetP:** Three red apples on a towel.

**LocalP:** Three red apples.

| Source image | Mask | CannyEdit | | Source image | Mask | CannyEdit |

Figure 16: Examples of source images, input masks and corresponding text prompts (SourceP: source prompt; TargetP: global target prompt; LocalP: local edit prompt) in RICE-Bench, along with CannyEdit's outputs.

## F  LIMITATIONS AND FUTURE IMPROVEMENTS

Compared to instruction-based editors like FLUX.1 Kontext and Qwen-Image-Edit, CannyEdit has a notable limitation: it cannot perform many free-form editing tasks (e.g., transformations that preserve a human subject's identity). Instead, CannyEdit's strengths lie in region-based editing tasks such as object addition, removal, and replacement. Nevertheless, CannyEdit goes beyond traditional region-based methods by supporting multiple edits in a single pass, incorporating point-based hints, and seamlessly integrating with VLM-inferred points for guidance. This difference highlights a trade-off between *flexibility and controllability*: instruction-based editors offer greater task flexibility, whereas CannyEdit provides finer control over where and how edits are applied, resulting in better preservation of the image's context. For open-ended creative editing, users might prefer instruction-based editors. In contrast, for professional editing tasks requiring precision—such as an interior designer adding specific objects at exact locations and sizes—CannyEdit is the better option due to its superior controllability.

To further improve controllability, we plan to introduce local control signals within the edit region in future work. Currently, CannyEdit relaxes its structural (Canny) control inside the edit region, meaning that portion of the image is guided mainly by the text prompt. We will explore injecting additional control inputs into this local area—for example, allowing a user to specify the pose of a person to add or the depth at which an object is inserted. By incorporating such fine-grained, local directives, CannyEdit could grant users even more precise control over complex edits.

Another potential limitation of CannyEdit is its reliance on a VLM to provide point hints for editing in an instruction-based setting. This dependency means the editing quality can be constrained by the accuracy of the VLM's inferred output. Fortunately, our approach allows for the training-free use of leading VLMs, a method we have proven to be effective. Nevertheless, we aim to further explore if and how we can activate the capabilities of weaker VLMs in combination with CannyEdit to achieve high-quality edits. One promising direction is the implementation of a two-stage training framework, beginning with Supervised Fine-Tuning (SFT) and followed by Reinforcement Learning (RL). This SFT-RL approach would be initiated using "warm-starting" data, which could be from the most advanced VLMs to provide a strong initial foundation for training their less powerful counterparts. We intend to explore this methodology in future work.

## G  REPRODUCIBILITY STATEMENT

We have provided technical details of CannyEdit in the main paper. To ensure the reproducibility of our experiments and to foster future research, we will release our code and curated benchmark upon the acceptance of the paper.

## H  ETHICS STATEMENT

The image editing technique introduced in this paper offers significant societal benefits. By providing intuitive tools for image modification, this approach reduces time and expense associated with editing tasks. This enhanced accessibility democratizes visual content creation, empowering users regardless of technical expertise.

We emphasize that our approach is a training-free method built on the FLUX.1 [dev] model, which avoids fine-tuning or custom training that could expand capabilities beyond the base model's intended scopes. Therefore, our method inherently relies on the existing safeguards of FLUX.1 [dev].

All referenced data sources and codebases are open-source. In alignment with open science principles, we will release our code and curated benchmark. License terms of all referenced resources will be strictly honored during release, ensuring full compliance.

For human evaluations, participants differentiated AI-edited images from real images using general daily-life samples requiring no expertise and containing no harmful content. Participants received compensation aligned with local standards.

## I  STATEMENT ON USAGE OF LLMS

The use of LLMs was focused on two distinct aspects, both detailed below, and their contribution did not extend to assuming authorship responsibilities or making intellectual contributions to the conclusions. Firstly, LLMs were employed to polish the descriptive text within certain paragraphs of the manuscript. This involved assisting with grammatical corrections, improving sentence flow, and enhancing the clarity of the writing. It is important to emphasize that all substantive academic content including the research questions, methodological design, analysis of results, and theoretical discussions, are originated solely from the authors. The LLM acted merely as a tool for language refinement, ensuring that the presentation of our original ideas was clear and professionally articulated.

Secondly, LLMs played a role in the construction of data samples, specifically in the context of manipulating the evaluation dataset, RICE-bench. Its curation is introduced in Appendix E. In this capacity, the model was used to generate the textual prompts that were subsequently employed to create images for evaluation and analysis. The LLM did not generate the final data or interpret the results; rather, it assisted in producing the initial structured linguistic inputs required for this specific technical process.

