# OpenReview forum: "CannyEdit: Selective Canny Control and Dual-Prompt Guidance for Training-free Image Editing"
_ICLR.cc/2026/Conference — ICLR 2026 Conference Withdrawn Submission_

### Official Review · Reviewer_63Ef · 2025-10-28

**Soundness:** 2
**Presentation:** 3
**Contribution:** 2
**Rating:** 4
**Confidence:** 4

**Summary:**

This paper addresses the fundamental trilemma in regional image editing - balancing text adherence, context fidelity, and editing seamlessness - by introducing CannyEdit, a novel training-free framework. The method features two key innovations: Selective Canny Control that applies structural guidance exclusively to unedited regions, preserving original details while enabling precise text-driven modifications; and Dual-Prompt Guidance that utilizes both local editing instructions and global scene descriptions to ensure accuracy and coherence. This approach supports flexible interactions ranging from rough masks to single-point hints, and seamlessly integrates with VLMs for complex instruction-based edits requiring planning and reasoning. Comprehensive evaluations demonstrate CannyEdit's superior performance in challenging object addition scenarios compared to state-of-the-art methods.

**Strengths:**

- The proposed editing method can precisely locate editing regions, support more flexible editing operations, and deliver highly faithful generation results.
- The designed approach effectively preserves the unedited regions of the image.
- The paper is clearly written, with well-presented comparative results and professionally crafted figures.

**Weaknesses:**

- The method appears to be heavily engineered, with the overall image generation process resembling a combination of null-text inversion, ControlNet, and FLUX.
- Given the involvement of multiple models, the inference speed requires clarification through detailed runtime analysis.
- The current strategy may struggle with editing requests involving significant spatial transformations, such as shifting the viewpoint by 60 degrees or making objects "fly" in the image. The authors should address how such limitations could be overcome.

**Questions:**

While current image editing models can handle complex instructions involving world knowledge and spatial transformations, and the paper compares with state-of-the-art models, these unified models offer comprehensive capabilities with user-friendly interfaces. I encourage the authors to further elaborate on what unique contributions their pipeline can provide in the context of rapidly advancing unified model research. I will determine my final score based on the authors' rebuttal and other reviewers' comments.

---

> ### Author Response · Authors · 2025-11-23
>
> Thanks for the review, we sincerely appreciate your feedbacks and insights towards our work. Regarding your concerns and questions, we response below.
>
> ### 1. "Given the involvement of multiple models, the inference speed requires clarification through detailed runtime analysis."
>
> We first report the runtime and GPU memory consumption of FLUX and FLUX + CannyEdit for the text-to-image (T2I) task.
>
> Next, we evaluate the performance of CannyEdit under the mask-based setting and compare it with KV-Edit. The official implementation of KV-Edit employs Vanilla ReFlow for inversion. To ensure a fair comparison with our CannyEdit, which utilizes FireFlow for inversion, we further report results of KV-Edit when equipped with FireFlow. The results show that CannyEdit (FireFlow) achieves comparable latency to KV-Edit (FireFlow) and is substantially faster than KV-Edit (Vanilla ReFlow). Although CannyEdit consumes more VRAM due to the incorporation of Canny ControlNet and dual prompt guidance, its memory footprint remains within an acceptable range for modern GPUs.
>
> We further report the runtime and GPU memory of CannyEdit under the instruction-based setting, where a VLM-inferred point hint is provided and CannyEdit automatically refines it into a mask using SAM-2 (introduced in Section 3.4). We compare this to FLUX-Kontext and Qwen-Edit (which is also provided with the VLM-inferred point hints), both of which are instruction-based and mask-free. The results show that, while the additional point-based refinement and SAM-2 processing make CannyEdit more memory-intensive, but it is still more efficient than the Qwen-Edit in terms of both VRAM used and latency.
>
> | **Method** | **Setting / Variant** | **VRAM (GB)** | **Latency (ms)** |
> |:------------|:----------------------|:-------------:|:----------------:|
> | **FLUX** | T2I baseline | 32.60 | 7,632.98 |
> | **FLUX + Canny ControlNet** | T2I with edge control | 34.08 | 8,008.73 |
> |||||
> | **CannyEdit** | Mask-based (inversion, FireFlow) | 37.96 | 13,828.97 |
> | **KV-Edit** | Mask-based (inversion, Vanilla ReFlow) | 32.75 | 33,876.73 |
> | **KV-Edit** | Mask-based (inversion, FireFlow) | 32.70 | 12,815.85 |
> |||||
> | **CannyEdit** | Mask-free (with provided point hint + SAM2 for  mask refinement) | 45.17 | 30,213.89 |
> | **FLUX-Kontext** | Mask-free (instruction-based) | 33.85 | 12,927.55 |
> | **Qwen-Edit** | Mask-free (instruction-based) | 58.08 | 36,715.37 |
>
> Note: The results are computed based on performing generation or edits on a (512, 512) image with 50 denoising steps using an H800 GPU.
>
> ### 2. Positions and unique contributions of CannyEdit in the context of rapidly advancing unified model research.
>
> We appreciate the reviewer’s insightful comments. Regarding the current limitations, we acknowledge that the tasks CannyEdit can handle primarily fall within the mask-based (inpainting) setting, such as object addition, replacement, and removal. Below, we clarify the positioning of CannyEdit and its unique contributions:
>
> -	Relative to prior mask-based/inpainting editors (e.g., KV-Edit, BrushEdit, FLUX Fill, and PowerPaint): CannyEdit achieves a superior balance among background preservation, text adherence, and editing seamlessness, as demonstrated in Table 1, Figures 3, 7, 8. Notably, unlike these methods, CannyEdit does not require explicit mask inputs. Instead, it can leverage VLM-inferred point hints to automatically refine masks, reducing the user’s input requirements while maintaining high-quality results.
>
> -	Relative to instruction-based editors (e.g., Qwen-Edit and FLUX-Kontext): With identical point hints, CannyEdit outperforms these methods in adhering to edit instructions, particularly in complex scenarios such as performing two edits in a single generation pass, as evidenced in Table 2, Figures 1 and 10. Additionally, CannyEdit offers stronger controllability over where the edits are applied. In Appendix B.3 and B.4, we explored various ways of incorporating mask information into instruction-based editors and observed that both FLUX-Kontext and Qwen-Edit struggled to localize edits effectively to user-specified regions.
>
> We will further clarify these points and elaborate on CannyEdit’s unique contributions in the context of rapidly advancing unified model research in the revised version of the paper.

---

### Official Review · Reviewer_fTcW · 2025-11-01

**Soundness:** 2
**Presentation:** 2
**Contribution:** 2
**Rating:** 2
**Confidence:** 4

**Summary:**

The paper introduces a training-free and model-agnostic approach for image editing. To preserve non-target pixels, the method adds a ControlNet model and injects masked ControlNet's features exclusively into unedited regions and guides edits with dual prompts (local for spatial precision, global for scene coherence). User point hints are translated by a VLM and refined into accurate masks, enabling a mask-based editing pipeline. The evaluation separates mask-based and instruction-only settings to ensure scope parity, and the appendix reports background-preservation metrics (PSNR/LPIPS/MSE) alongside Context Fidelity.

**Strengths:**

- The method requires no fine-tuning of the base diffusion model (i.e., FLUX.1-dev) and works with existing Canny-based ControlNets, making integration straightforward and model-agnostic.
- During denoising, Canny ControlNet feature maps are injected only into non-target pixels, stabilizing layout and preventing unintended changes in unedited regions.
- Dual (local/global) prompts and a VLM+SAM pipeline that converts point hints into accurate masks enable precise control and natural extension to a wide range of editing tasks.

**Weaknesses:**

**1. Minor Novelty of CannyEdit.** Despite using provided editing masks (or VLM+SAM–refined masks), background preservation lags far behind prior art: e.g., LPIPS (Appendix Tab. 4) shows KV-Edit 9.92 vs. CannyEdit 26.38, indicating that simply mixing ControlNet features into non-target pixels is insufficient to protect unedited regions. If the method’s core claim is “Selective Canny Control preserves the original structure,” then background-fidelity metrics must be strong in the main tables; moving them to the appendix while underperforming undermines the central contribution. In addition, comparisons to training-free background-preservation baselines that do not rely on masks (e.g., [1]) are limited and should be expanded. Also, the proposed dual-prompt guidance aligns with techniques widely used in layout-to-image generation [2–5]. Beyond direct application to editing, the paper should clarify what is technically new (e.g., objective, optimization, or inference mechanism) and why it matters for editing beyond prior formulations.

**2. Limited Experiments.** Although the approach should be compatible with any ControlNet-augmented UNet, experiments focus on FLUX.1-[dev] + FLUX-Canny-ControlNet only. The paper should delineate scope and limits, and test other controllable structures (e.g., GLIGEN [6], ControlNeXt [7]) where Canny/edge cues can serve as conditions. In Appendix C.4 (ControlNet Strength), background preservation should be evaluated with standard metrics (PSNR/SSIM/LPIPS refer to KV-Edit and [1]) on unedited regions to quantify whether increasing “strength” truly keeps non-targets intact. If the goal is “structure preservation,” it is natural to assess other edge/depth priors (e.g., HED, depth maps) and why Canny is preferred. The current ablations do not sufficiently justify this choice. Because ControlNet runs alongside the base model, compute and memory roughly double; the paper should report VRAM/latency vs. quality trade-offs to argue for practical value.

**3. Unclear Setup and Unfair Comparison.** Masks appear mandatory for the Selective Canny Control pipeline, yet several figures (e.g., Figure 6) are ambiguous about which mask (provided vs. refined) was used. Since Kontext/Qwen-Edit do not take masks, the paper should center comparisons within mask-based editing to avoid conflating settings and to attribute gains fairly. Background-preservation metrics must appear in the main tables (e.g., Table 1), not only in the Appendix. The primary claim is preservation of unedited regions; readers need headline numbers against strong baselines to gauge effectiveness. Hiding weaker numbers in the Appendix invites doubts about the method’s core contribution.

[1] Early Timestep Zero-Shot Candidate Selection for Instruction-Guided Image Editing, ICCV 2025.

[2] Region-Aware Text-to-Image Generation via Hard Binding and Soft Refinement, ICCV 2025.

[3] DreamRenderer: Taming Multi-Instance Attribute Control in Large-Scale Text-to-Image Models, ICCV 2025.

[4] NoiseCollage: A Layout-Aware Text-to-Image Diffusion Model Based on Noise Cropping and Merging, CVPR 2024.

[5] GrounDiT: Grounding Diffusion Transformers via Noisy Patch Transplantation, NeurIPS 2024.

[6] GLIGEN: Open-Set Grounded Text-to-Image Generation, CVPR 2023.

[7] ControlNeXt: Powerful and Efficient Control for Image and Video Generation

**Questions:**

Q1. How does Selective Canny Control differ technically from simply mixing ControlNet features, and why should this yield superior background preservation (i.e., the original image's details)? Given this is the paper’s central claim, why are background-preservation metrics absent from the main tables (Tab. 1/2)? For instance, Appendix Tab. 4 reports LPIPS 9.92 (KV-Edit) vs. 26.38 (CannyEdit) under identical masks/inputs. What explains this gap, and can you provide failure cases (qualitative examples and per-region metrics) diagnosing where Selective Canny fails?

Q2. Dual-prompt guidance appears close to prior layout-to-image controls [2–5]. What is the editing-specific novelty beyond a simple adaptation, and can you ablate to show gains that prior prompting schemes cannot match (e.g., with/without dual prompts, prompt-mixing strategies, or attention routing ablations)?

Q3. Results focus on FLUX.1-[dev] + FLUX-Canny. To clarify scope and limits, can you report on additional control frameworks (e.g., classic U-Net + ControlNet, GLIGEN [6], ControlNeXt [7])? If the goal is structural fidelity, can you also test other priors (e.g., HED edges, depth) or provide evidence that Canny is the preferred/most stable signal?

Q4. What are the compute/memory overheads (e.g., VRAM and Latency) of running ControlNet alongside the base model? Please include a quality vs. cost table (background-preservation and perceptual metrics vs. FLOPs/VRAM/Latency) to substantiate practical viability.

---

> ### Author Response · Authors · 2025-11-23
> **Rebuttal (1/3)**
>
> Thank you for the review; we sincerely appreciate your feedback and insights on our work. Regarding your concerns and questions, our responses are provided below.
>
> ### 1. "How does Selective Canny Control differ technically from simply mixing ControlNet features, and why should this yield superior background preservation (i.e., the original image's details)?"
>
> First, we clarify that **we did not claim Selective Canny Control yields superior background preservation.** *CannyEdit is designed to strike a balanced trade-off among background preservation, text adherence, and editing seamlessness.* If one desires 100% background preservation, a trivial approach is to copy the original background pixels directly. KV-Edit replaces the edited image tokens with those from the original image during denoising; while this achieves strong background preservation, it significantly harms editing seamlessness (see visual examples in Figure 3 and Figure 9 and the lower Perceptual Realism in Table 1).
>
> In contrast, CannyEdit performs preservation and editing in a “softer” way. Recall that a Canny ControlNet provides optional conditioning to the base text-to-image model: when ControlNet outputs are present, generation leverages both the Canny map and the text; when absent, generation follows the text alone. Selective Canny Control exploits this property by spatially gating the ControlNet conditioning: we inject cached ControlNet’s features (obtained during inversion) only in the background region to preserve context, while withholding ControlNet features in the edit region so its content is driven purely by text—using both the local and global prompts under our dual-prompt guidance.
>
> As a result, Selective Canny Control does not pursue maximal background preservation at all costs, but a controlled balance: strong context preservation in the background without structurally constraining the edited region, thereby improving edit quality and realism.
>
>
> ### 2. Regarding evaluation results on the background preservation.
>
> Your observation regarding the LPIPS scores is correct. However, KV-Edit’s superior score in this metric comes at a significant cost to editing quality and realism, resulting in noticeable artifacts and a lack of seamless integration, as we demonstrate in Figures 3 and 9. Figure 9 specifically includes examples where CannyEdit’s LPIPS is higher, yet the visual result is more coherent. We do not classify these as failures in background preservation, but rather as a deliberate trade-off.
>
> This behavior is an intentional choice rooted in the design of our attention mechanism among image tokens. As detailed in Equation (6), we allow background areas immediately adjacent to the edit boundary to attend to the edited region's features. This enables the model to create a more natural and seamless transition by making subtle adjustments to the boundary context. While this process may slightly alter pixels in the non-masked region and thus increase the LPIPS score, it is crucial for achieving the high degree of editing seamlessness that characterizes our method.
>
> We fully agree that the background-preservation metrics are vital for a comprehensive comparison and should be featured more prominently. We will move these metrics from the appendix to the main tables in the revised version of the paper and expand our discussion to clarify this important trade-off.

---

> ### Author Response · Authors · 2025-11-23
> **Rebuttal (2/3)**
>
> ### 3. The relationship between our proposed dual-prompt guidance and the techniques used in layout-to-image generation
>
> a. Compared to [1,2], our dual-prompt guidance uses a single-pass, unified attention mechanism that jointly handles local and global prompts. This is more efficient and coherent than the two-stage pipelines in [1,2], which generate with local and global prompts separately and then blend latents. Our unified design is empirically superior for editing: replacing it with the latent blending strategy of [1] yields a clear drop in Context Fidelity (88.72 in Table 1 *add* → 84.69) and Text Adherence (28.12 → 24.85), validating the effectiveness of single-pass unification. As for [3], although a “global” prompt is mentioned, it is effectively applied as a local prompt to the background, not as a true global guidance signal.
>
> b. Relative to [4], our attention flow is tailored for editing. The edit region can attend to the background for natural integration, and to itself plus the local prompt for strong text adherence. For the background, we mask image attention flow between background and edit region to prevent unintended contextual drift. However, as specified in Equation (6), background areas adjacent to the edit boundary are allowed to attend to the edit region, enabling them to absorb cues from the edited content and thereby improving boundary seamlessness.
>
> [1] Training-free Regional Prompting for Diffusion Transformers, arxiv.
>
> [2] Region-Aware Text-to-Image Generation via Hard Binding and Soft Refinement, ICCV 2025.
>
> [3] NoiseCollage: A Layout-Aware Text-to-Image Diffusion Model Based on Noise Cropping and Merging, CVPR 2024.
>
> [4] DreamRenderer: Taming Multi-Instance Attribute Control in Large-Scale Text-to-Image Models, ICCV 2025.
>
> ### 4. "Masks appear mandatory for the Selective Canny Control pipeline, yet several figures (e.g., Figure 6) are ambiguous about which mask (provided vs. refined) was used. Since Kontext/Qwen-Edit do not take masks, the paper should center comparisons within mask-based editing to avoid conflating settings and to attribute gains fairly."
>
> We appreciate the reviewer’s point and will clarify in the figure captions that masks are used. For Figure 6 specifically, the masks are automatically refined from VLM-inferred points following the point-inference procedure described in Appendix B.5.1.
>
> We also note that **while masks are mandatory for the Selective Canny Control pipeline, masks are not mandatory for CannyEdit**. CannyEdit supports either user-provided masks or masks automatically refined from a user-provided/VLM-inferred point hint (see Section 3.4). In our evaluations, we aimed for fair comparisons: when comparing against mask-based editors, we use the same masks; when comparing against instruction-based editors, we provide the same VLM-inferred point hints. **Under this setup, comparisons to FLUX-Kontext and Qwen-Edit are fair because all these methods do not take user-provided masks; instead, we supply identical VLM-inferred point hints, and our masks are automatically refined from those hints.** We further explored incorporating mask information into these instruction-based editors (Appendix B.3 and B.4) and observed that FLUX-Kontext and Qwen-Edit struggled to localize edits to the specified regions.
>
> Our positioning of CannyEdit is as follows:
>
> a. Relative to prior mask-based editors, CannyEdit achieves a better balance among background preservation, text adherence, and editing seamlessness.
>
> b. Relative to instruction-based editors, CannyEdit offers stronger controllability over “where to edit,” and better follows the text prompt in complex scenarios (e.g., Figure 1(a)(b)).
>
> For these reasons, we believe both parts of the experimental comparison are important and complementary.
>
> ### 5. "Although the approach should be compatible with any ControlNet-augmented UNet, experiment results focus on FLUX.1-[dev] + FLUX-Canny... If the goal is structural fidelity, can you also test other priors (e.g., HED edges, depth) or provide evidence that Canny is the preferred/most stable signal?"
>
> Regarding the claim that the approach should be compatible with any ControlNet-augmented UNet, please note that we did not make this claim. The setup in the paper is specifically based on the diffusion transformer framework. We will clarify the scope and limitations in the revised version of the paper to avoid confusion.
>
> As for testing other priors (e.g., HED edges, depth), a detailed comparison between Canny ControlNet and HED/Depth ControlNet is provided in Appendix C.3. We selected Canny to prioritize the primary objectives of mask-based editing: faithfully preserving background context while enabling strong generative flexibility within the edited region, rather than focusing solely on structural fidelity. The results in Appendix C.3 show that Canny achieves the best balance between context preservation and instruction adherence within our framework.

---

> ### Author Response · Authors · 2025-11-23
> **Rebuttal (3/3)**
>
> ### 6.	"Because ControlNet runs alongside the base model, compute and memory roughly double; the paper should report VRAM/latency vs. quality trade-offs to argue for practical value. What are the compute/memory overheads (e.g., VRAM and Latency) of running ControlNet alongside the base model? "
>
> “Because ControlNet runs alongside the base model, compute and memory roughly double.” **This statement is not correct.** As clarified in Lines 200–203 of the paper, *“The FLUX-Canny-ControlNet integrates duplicates of two multistream blocks from FLUX to inject structural layout guidance into it,”* while there are 19 multistream blocks and 38 single-stream blocks in total. Thus, the computation and memory overhead are far less than double.
>
> Below, we report the runtime and GPU memory consumption of FLUX and FLUX + CannyEdit for the text-to-image (T2I) task. Then, we evaluate the efficiency performance of CannyEdit in the mask-based setting and compare it with KV-Edit. The official implementation of KV-Edit uses Vanilla ReFlow for inversion, but to ensure a fair comparison with our CannyEdit (which uses FireFlow for inversion), we also report results for KV-Edit equipped with FireFlow.
>
> The results demonstrate that CannyEdit (FireFlow) achieves comparable latency to KV-Edit (FireFlow) and is significantly faster than KV-Edit (Vanilla ReFlow). Although CannyEdit consumes slightly more VRAM due to the inclusion of Canny ControlNet and dual-prompt guidance, its memory footprint remains within acceptable limits for modern GPUs.
>
> | **Method** | **Setting / Variant** | **VRAM (GB)** | **Latency (ms)** |
> |:------------|:----------------------|:-------------:|:----------------:|
> | **FLUX** | T2I baseline | 32.60 | 7,632.98 |
> | **FLUX + Canny ControlNet** | T2I with edge control | 34.08 | 8,008.73 |
> |||||
> | **CannyEdit** | Mask-based (inversion, FireFlow) | 37.96 | 13,828.97 |
> | **KV-Edit** | Mask-based (inversion, Vanilla ReFlow) | 32.75 | 33,876.73 |
> | **KV-Edit** | Mask-based (inversion, FireFlow) | 32.70 | 12,815.85 |
>
> Note: The results are computed based on performing generation or edits on a (512, 512) image with 50 denoising steps using an H800 GPU.
>
> ### 7. Quality vs. Cost Analysis.
>
> We conducted the quality vs. cost analysis using the settings in Table 1 (add), adding objects with given masks. Since the cost is only related to the denoising step in our CannyEdit method, we report the context fidelity and text adherence performance with respect to different denoising steps below. Our results show that even when reducing the workload to 30 or 40 steps, our method's Text Adherence (25.98 and 26.62, respectively) remains significantly higher than both the KV-Edit (17.25) and PowerPaint-FLUX (24.34) baselines. A more significant performance drop in text adherence is only observed at 20 steps, though the context fidelity is slightly better.
>
> | Steps | Context Fidelity ↑ | Text Adherence ↑ |
> |-----------------|---------------------|------------------|
> | 20 | 90.12 | 23.58 |
> | 30 | 89.12 | 25.98 |
> | 40 | 88.51 | 26.62 |
> | 50 (default) | 88.72 | 28.12 |
> | KV-Edit | 93.91 | 17.25 |
> | PowerPaint-FLUX | 84.63 | 24.34 |

---

> ### Comment · Reviewer_fTcW · 2025-11-26
>
> I deeply appreciate the authors for their detailed rebuttal and the additional experiments provided. These have helped clarify several misunderstandings, particularly regarding compute overhead, and improved the transparency of the evaluation, such as the clarifications for Figure 6. However, while some logistical questions have been resolved, fundamental issues regarding the method's preservation capability and experimental settings remain.
>
> * Regarding background preservation and the claimed "deliberate trade-off", I remain unconvinced. The "boundary attention" mechanism effectively explains local blending but fails to justify the global color shifts observed in Figure 3 or the significant LPIPS gap (9.92 for baseline vs. 26.38 for CannyEdit). This degree of deviation suggests a system-wide re-generation rather than preservation. Furthermore, the unpredictable mask adherence seen in Figure 9, where objects often ignore user-specified constraints, implies that the method treats user inputs as loose suggestions. If the paper intends to keep these results, the revised version must explicitly define the method’s scope, clarifying that it prioritizes generative seamlessness over pixel-level fidelity and strict mask adherence.
>
> * On the topics of fairness and novelty, I appreciate the clarifications on the single-pass attention design. However, a methodological gap persists in the comparisons: CannyEdit leverages an external model (SAM) to refine inputs as described in Section 3.4, whereas baselines do not. The revised paper should explicitly state that the localization advantage stems partly from this external supervision pipeline. Additionally, while the single-pass architecture is distinct, the claim that it "prevents unintended contextual drift" is contradicted by the empirical evidence of global color shifts and high LPIPS scores discussed above.
>
> * Regarding the scope and generality discussed in your response (5), I must correct the interpretation of my previous comment. While I understand the current implementation utilizes a DiT-based framework (FLUX), the core mechanism of selectively injecting masked ControlNet features is fundamentally architecture-agnostic. Therefore, my suggestion to test broader applicability on U-Net architectures was a valid inquiry into the method's potential generality rather than a misunderstanding of the current setup.
>
> * Regarding compute and experimental settings, I stand corrected on the "doubling compute" assumption and thank the authors for the VRAM/latency data. However, I am concerned that the benchmarks were conducted at 512x512 resolution. Since FLUX.1 is optimized for 1024x1024, running at a significantly lower resolution involves a latent size four times smaller than intended, potentially masking artifacts or quality issues. Additionally, the non-monotonic trends in the step ablation study raise questions about stability and whether high fidelity at low steps simply reflects incomplete inversion.
>
> Major concerns still remain regarding the integrity of unedited regions, the precision of mask adherence, and the validity of evaluating at non-native resolutions. Therefore, I currently maintain my rating, but I remain open to adjusting my score should the authors provide compelling evidence addressing my concerns.

---

### Official Review · Reviewer_jng2 · 2025-11-01

**Soundness:** 3
**Presentation:** 3
**Contribution:** 3
**Rating:** 4
**Confidence:** 4

**Summary:**

This paper presents **CannyEdit**, a *training-free* image editing framework designed to balance **editability and fidelity**. The method introduces two key components: **Selective Canny Control**, which preserves unedited structures using edge-based guidance, and **Dual-Prompt Guidance**, which combines local and global text prompts for coherent editing. A progressive mask refinement strategy further supports weak or point-based editing inputs. Experiments on **RICE-Bench** show that CannyEdit achieves superior text alignment and visual realism compared to recent baselines such as KV-Edit, BrushEdit, PowerPaint-FLUX, FLUX.Fill.

**Strengths:**

### Originality
1. Introduces a clear and well-motivated training-free editing framework addressing the editability-fidelity-seamlessness trade-off.
2. Overall originality is moderate: innovation lies more in integration and careful design than in novel algorithms.

### Quality
1. Strong empirical validation on both mask-based and instruction-based setups.
2. Selective Canny Control effectively preserves background structure while allowing flexible local edits.
3. Dual-Prompt Guidance improves text alignment and global coherence compared with single-prompt baselines.
4. The method supports weak or point-based inputs, enabling integration with VLM-driven editing.

### Significance
1. Offers a practical, flexible editing solution that bridges mask-based and instruction-based paradigms.
2. The framework’s ability to use VLM-inferred point hints makes it relevant for future reasoning-based editing systems.

**Weaknesses:**

1. Limited Novelty of Mechanisms
The two key modules—selective structural control and multi-prompt attention—mainly extend existing ControlNet and attention-masking strategies rather than introducing fundamentally new formulations. In particular, **Selective Canny Control** is conceptually similar to the edge-based structural guidance used in **MagicQuill[1] (Sec. 3.1, Para. 1)**, but the paper does not explicitly clarify how it differs.

2. Efficiency Unclear
Although the method is “training-free,” it still involves inversion, ControlNet caching, and multi-prompt attention, all of which can be computationally expensive. The paper does not report quantitative runtime or memory comparisons against strong training-free/editing baselines such as KV-Edit or PowerPaint-FLUX, and the comparison with Qwen-Image-Edit in the appendix is not sufficient to assess practical efficiency — a more complete evaluation is needed.

3. Clarity and Structure Issues

The **Method** section feels overly long and dense, which makes it hard to follow. Some parts — for example, *Dual-Prompt Guidance* — would be much clearer with an attention diagram instead of long text descriptions, while the more basic *Preliminaries* could easily be moved to the appendix.

Because method section runs almost to page 7, many important experiments and analyses (like the ablation studies) are pushed into the appendix, leaving the main paper incomplete. Ideally, the core technical and experimental content should appear in the main text, with the appendix used only for supporting details.

[1] Liu, Zichen, et al. "Magicquill: An intelligent interactive image editing system." Proceedings of the Computer Vision and Pattern Recognition Conference. 2025.

**Questions:**

1. What is the actual inference cost (time and GPU memory) compared to KV-Edit or PowerPaint-FLUX?

2. The proposed Selective Canny Control appears conceptually similar to the edge-based structural guidance in MagicQuill [1] (Sec. 3.1, Para. 1). Could the authors explicitly clarify the key differences in formulation or implementation, and explain how CannyEdit advances beyond prior edge-guided editing methods?

3. CannyEdit relies heavily on Canny-based structural guidance, which may underperform in texture-rich or low-edge regions (e.g., skies or artistic styles). How does the method handle such cases? How strong is the generalization of this method?

---

> ### Author Response · Authors · 2025-11-23
>
> Thank you for your review — we sincerely appreciate your feedback and the insights you've shared about our work. In response to your concerns and questions, please find our replies below.
>
> ## 1. Similarity between Selective Canny Control and the edge-based structural guidance used in MagicQuill.
>
> Conceptually, our selective Canny control shares similarities with the edge-based structural guidance used in MagicQuill.
>
> However, the two differ technically. Specifically, our approach applies selective control to the *cached Canny ControlNet outputs*—that is, at the latent-space level—obtained from the inversion process, whereas MagicQuill performs its edge subtraction directly on the *edge map*, at the input level.
>
> *In our design, both background preservation and editing flexibility within the target region are achieved through a single module: the Canny ControlNet.* The cached ControlNet outputs are integrated into the base text-to-image model to maintain background consistency, while the outputs within the editing region are masked to enable flexible edits. In contrast, MagicQuill removes edge information in the edit region of the edge map before fitting to the edge ControlNet, allowing editing in those regions, but relies on a *separate inpainting model* to read the original image and preserve the background. This inpainting model must be additionally trained if a suitable pre-trained version is unavailable.
>
> Thank you for highlighting this connection—we will cite and discuss MagicQuill in the revised version of the paper.
>
> ## 2. Analysis of run time and GPU memory
>
> We first report the runtime and GPU memory consumption of FLUX and FLUX + CannyEdit for the text-to-image (T2I) task.
>
> Next, we evaluate the performance of CannyEdit under the mask-based setting and compare it with KV-Edit. The official implementation of KV-Edit employs Vanilla ReFlow for inversion. To ensure a fair comparison with our CannyEdit, which utilizes FireFlow for inversion, we further report results of KV-Edit when equipped with FireFlow. The results show that CannyEdit (FireFlow) achieves comparable latency to KV-Edit (FireFlow) and is substantially faster than KV-Edit (Vanilla ReFlow). Although CannyEdit consumes more VRAM due to the incorporation of Canny ControlNet and dual prompt guidance, its memory footprint remains within an acceptable range for modern GPUs.
>
> We further report the runtime and GPU memory of CannyEdit under the instruction-based setting, where a VLM-inferred point hint is provided and CannyEdit automatically refines it into a mask using SAM-2 (introduced in Section 3.4). We compare this to FLUX-Kontext and Qwen-Edit (which is also provided with the VLM-inferred points), both of which are instruction-based and mask-free. The results show that, while the additional point-based refinement and SAM-2 processing make CannyEdit more memory-intensive, but it is still more efficient than the Qwen-Edit in terms of both VRAM used and latency.
>
> | **Method** | **Setting / Variant** | **VRAM (GB)** | **Latency (ms)** |
> |:------------|:----------------------|:-------------:|:----------------:|
> | **FLUX** | T2I baseline | 32.60 | 7,632.98 |
> | **FLUX + Canny ControlNet** | T2I with edge control | 34.08 | 8,008.73 |
> |||||
> | **CannyEdit** | Mask-based (inversion, FireFlow) | 37.96 | 13,828.97 |
> | **KV-Edit** | Mask-based (inversion, Vanilla ReFlow) | 32.75 | 33,876.73 |
> | **KV-Edit** | Mask-based (inversion, FireFlow) | 32.70 | 12,815.85 |
> |||||
> | **CannyEdit** | Mask-free (with provided point hint + SAM2 for  mask refinement) | 45.17 | 30,213.89 |
> | **FLUX-Kontext** | Mask-free (instruction-based) | 33.85 | 12,927.55 |
> | **Qwen-Edit** | Mask-free (instruction-based) | 58.08 | 36,715.37 |
>
> Note: The results are computed based on performing generation or edits on a (512, 512) image with 50 denoising steps using an H800 GPU.
>
> ## 3. Regarding your suggestion about the paper organization.
>
> Thanks for the insightful feedback. In the revised version of the paper, we streamline the Method section to improve clarity and readability, including adding an attention diagram for Dual-Prompt Guidance and moving basic preliminaries to the appendix as you suggested. Additionally, we will bring key results, such as the ablation studies, into the main paper to ensure it presents a more complete narrative.

---

> > ### Comment · Reviewer_jng2 · 2025-11-26
> >
> > Thank you for the detailed rebuttal and clarifications.
> >
> > Regarding the inference speed, CannyEdit does not demonstrate a clear advantage; in fact, it appears to be more than twice slower than FLUX-Kontext. This inference overhead substantially limits the practical contribution of the method. In addition, although the authors mentioned that updates would be included in the revised version, I could not identify any concrete modifications in the submitted PDF.
> >
> > Overall, the limited novelty, the current writing quality, and the relatively slow inference time constrain the impact of the work. Therefore, I will maintain my score leaning toward rejection.

---

### Note · Authors · 2025-12-01

I have read and agree with the venue's withdrawal policy on behalf of myself and my co-authors.